**Water and carbon stable isotope records from natural archives : a new**
**database and interactive online platform for data browsing, visualizing and**
**downloading.**
Timothé Bolliet[1], Patrick Brockmann[1], Valérie Masson-Delmotte[1], Franck Bassinot[1],
Valérie Daux[1], Dominique Genty[1], Amaelle Landais[1], Marlène Lavrieux[2], Elisabeth
Michel[1], Pablo Ortega[3], Camille Risi[4], Didier M. Roche[1], Françoise Vimeux[1,5], Claire
Waelbroeck[1].
1 - Institut Pierre Simon Laplace / Laboratoire des Sciences du Climat et de
l'Environnement, LSCE/IPSL, CEA-CNRS-UVSQ, Université Paris-Saclay, F-91191
Gif-sur-Yvette, France.
2 - Eawag, Swiss Federal Institute of Aquatic Science and Technology,
Überlandstrasse 133, 8600 Dübendorf, Switzerland.
3 - Laboratoire d'Océanographie et du Climat : Expérimentations et Approches
Numériques (LOCEAN) Université Pierre et Marie Curie, 4 Place Jussieu, 75252
Paris, France.
4 - Laboratoire de Météorologie Dynamique (LMD), place Jussieu, 75252 Paris
Cedex 05, France.
5- Institut de Recherche pour le Développement (IRD), Laboratoire HydroSciences
Montpellier (HSM), UMR 5569 (CNRS-IRD-UM1-UM2), 34095 Montpellier, France.
Correspondence to: T. Bolliet (Timothe.Bolliet@lsce.ipsl.fr)
**Abstract**
Past climate is an important benchmark to assess the ability of climate models to
simulate key processes and feedbacks. Numerous proxy records exist for stable
isotopes of water and/or carbon, which are also implemented inside the components
of a growing number of Earth system model. Model-data comparisons can help to

constrain the uncertainties associated with transfer functions. This motivates the need of producing a comprehensive compilation of different proxy sources. We have put together a global database of proxy records of oxygen ($\delta^{18}O$), hydrogen ($\delta D$) and carbon ($\delta^{13}C$) stable isotopes from different archives: ocean and lake sediments, corals, ice cores, speleothems and tree-ring cellulose. Source records were obtained from the georeferenced open access PANGAEA and NOAA libraries, complemented by additional data obtained from a literature survey. About 3,000 source records were screened for chronological information and temporal resolution of proxy records. Altogether, this database consists of hundreds of dated $\delta^{18}O$, $\delta^{13}C$ and $\delta D$ records in a standardized simple text format, complemented with a metadata Excel catalog. A quality control flag was implemented to describe age markers and inform on chronological uncertainty. This compilation effort highlights the need to homogenize and structure the format of datasets and chronological information, and enhance the distribution of published datasets that are currently highly-fragmented and scattered. We also provide an online portal based on the records included in this database with an intuitive and interactive platform (http://climateproxiesfinder.ipsl.fr/), allowing one to easily select, visualize and download subsets of the homogeneously-formatted records that conform this database, following a choice of search criteria, and to upload new datasets. In the last part, we illustrate the type of application allowed by our database by comparing several key periods highly investigated by the palaeoclimate community. For coherency with the Paleoclimate Modelling Intercomparison Project (PMIP), we focus on records spanning the past 200 years, the mid-Holocene (MH, 5.5-6.5 ka; calendar kilo years before 1950), and the Last Glacial Maximum (LGM, 19-23 ka), and those spanning the last interglacial period (LIG, 115-130 ka). Basic statistics have been applied to characterize anomalies between these different periods. Most changes from the MH to present day, and LIG to MH appear statistically insignificant. Significant global differences are reported from LGM to MH with regional discrepancies in signals from different archives and complex patterns.

## 1. Introduction

In the context of increasing anthropogenic greenhouse gas emissions, exploring future climate change risks relies on climate models (IPCC AR5, 2013), and it

becomes essential to assess their intrinsic skills and limitations (Braconnot et al.,
2012; Flato et al., 2013).
Past climate variations resulted from the changing natural external forcings, and
internal climate variability. Quantitative records of past climate variations therefore
provide unique benchmarks against which is it possible to assess the ability of
climate models to resolve the processes at play (e.g. Braconnot et al., 2012, Schmidt
et al., 2014). However, evaluating climate models against paleoclimate data remains
challenging, due to uncertainties on both simulations and reconstructions (Masson-
Delmotte et al., 2013; Flato et al., 2013). On the one hand, uncertainties associated
with the simulation of past climates are related to changes in boundary conditions
(e.g. ice sheet topography and melt fluxes, https://pmip3.lsce.ipsl.fr/) and dust
radiative feedbacks (Rohling et al., 2012). On the other hand, uncertainties also arise
from the age scales of proxy records, and from the application of transfer functions
used to convert proxy records into climate variables. For instance, while $\delta^{18}$O is used
as a temperature proxy in polar ice cores, the relationship between ice core $\delta^{18}$O and
temperature is known to vary trough time and between drilling sites (Masson-
Delmotte et al., 2011a; Guillevic et al., 2013; Buizert et al., 2014). Similarly, the
relationship between $\delta^{18}$O from tree rings cellulose and climate may be impacted by
several factors, including local monthly or annual temperature and precipitation, while
the response of trees to climate changes may differ according to inherent
physiological differences of the various tree species (Stuiver and Braziunas, 1987;
McCarroll and Loader, 2004).
In order to constrain the second source of uncertainty, a growing number of
components of climate models are being implemented with the explicit simulation of
tracers such as water and carbon stable isotopes. Since the pioneer work of
Joussaume et al. (1984), many models are being equipped with $\delta^{18}$O, $\delta$D and also
$\delta^{17}$O water isotopes, including land surface models (Yoshimura et al., 2006;
Henderson-Sellers et al., 2006), regional atmospheric models (Sturm et al., 2010)
general circulation models (Schmidt et al., 2007 for the coupled ocean-atmosphere
GISS model; Lee et al., 2008 for NCAR CAM2; Tindall et al., 2009 for HadCM3; Risi
et al., 2010 for LMDZ4; Werner et al., 2011 for ECHAM5wiso; Yoshimura et al., 2011
for IsoGSM; Dee et al., 2015) as well as intermediate complexity climate models
(Roche et al., 2013 for iLOVECLIM). Similarly, carbon stable isotopes are also
implemented in a growing number of land surface and ocean components (e.g.
Tagliabue et al., 2009; Menviel et al., 2012; Sternberg et al., 2009). These new
functionalities of climate models open the possibility to directly comparing the proxies
measured in natural archives with model output, with the double interest of improving
the understanding of proxy records, and model evaluation. For instance, Risi et al.
(2010) evaluated LMDZ4 performance against oxygen stable isotope data from
terrestrial and ice archives for the MH and LGM, and Oppo et al. (2007) compared
the GISS Model-E output with Pacific marine $\delta^{18}O$ records encompassing the MH.
Recently, Caley and Roche (2013) have focused on the difference between the LGM
and the Late Holocene (last 1000 years) for the comparison of the simulation from
the iLOVECLIM model and proxy data, and selected 17 polar ice core records, 10
speleothems, and 116 deep sea cores with a test on age control following the
protocol previously applied for the synthesis of temperature reconstructions by the
Multiproxy Approach for the Reconstruction of the Glacial Ocean surface (MARGO)
collaborative effort (Waelbroeck et al., 2009). Also, Jasechko et al. (2015) compiled
88 isotope records from ground water, speleothems and ice cores spanning the
period from the LGM to the Late Holocene and compared these data to five general
circulation models. These model-data comparisons have only used limited
information extracted from a fraction of available proxy records, while much broader
information has been accumulated during decades of field and laboratory work
worldwide.
The main open-access databases are hosted on the NOAA
(http://www.ncdc.noaa.gov/data-access/paleoclimatology-data) and PANGAEA
(http://www.pangaea.de) websites. These multi-proxy online data depositories are
continuously updated with recent datasets uploaded by the respective authors on a
voluntary basis. In some cases, datasets are also available as supplementary
information to publications, and practices depend on communities. For instance,
there is no standard practice for archiving the growing number of stable isotope
records obtained from tree ring cellulose, even though some efforts emerged recently
to create a data bank (Csank, 2009). Although the two repositories have been
intensively used by scientists to archive and distribute their datasets, the systematic
exploration of these records remained limited by the heterogeneity of reporting, data
formats including chronological information, and the impossibility to easily download
all the datasets related to one type of proxy. Moreover, these databases have limited
interactivity. The lack of features allowing an online pre-visualization of selected
datasets obliges the users to download the data if they want to assess the relevance
of the records for their scientific questions (e.g. to explore the resolution of the
records, or the quality of the chronology for a given time interval). Altogether,
unintuitive ergonomics and/or limited interactivity make data browsing and gathering
fastidious.
Based on this observation, we decided to produce a compilation of existing records,
standardising the chronological information (age markers) into a common format, and
implementing an online tool to facilitate the search process throughout different
archives with intuitive data browsing, online functions for datasets graphical pre-
visualization, as well as easy download features. In a first step, we focus here on
$\delta^{18}O$, $\delta D$ and, if available on the same archive, $\delta^{17}O$ and $\delta^{13}C$. This choice is
motivated by the following reasons: (i) these proxies have been widely used during
the last decades; (ii) they are available for a variety of marine, ice and terrestrial
archives (sediments, speleothems, ice and tree-ring cellulose), and (iii) they trace
interactions between different components of the climate system involved in the
global water and carbon cycles, and provide therefore integrated signals for
evaluating respectively water and carbon cycle processes within climate simulations.
A strong motivation for this compilation is the integration of marine and terrestrial
records (Bar-Matthews et al., 2003; Hughen et al., 2006; Cruz et al., 2006; Leduc et
al., 2009; Carré et al., 2012; Bard et al., 2013; Grant et al., 2012 & 2014). It is also in
line with ongoing efforts to build consistent chronologies for marine and ice core
records (e.g. the INTIMATE project, see Blockley et al., 2012). In order to document
the four dimensional structure of ocean circulation changes, we included datasets
from deep-sea sediments, using both surface and deep water proxies.
While in principle our methodology could allow one to explore transient climatic
changes (Marcott et al., 2013; Shakun et al., 2012), such an approach would require
an accurate assessment of age scale uncertainties, which is beyond the scope of this
work. In this manuscript, we therefore focus on records providing sufficient age
control and resolution for selected time slices, chosen for consistency with the
Paleoclimate Modelling Intercomparison Project (PMIP), and for which numerous

source records are available. The selection of target periods is described in Section 2. The protocols and methods used to build the database are then depicted in Section 3, followed by the description of the software developments required for the online search and visualization platform (Section 4). For the four considered time slices, we then illustrate the data coverage and spatial distributions (Section 5). Conclusions provide recommendations to facilitate such data syntheses, and propose future database developments.

## 2. Selection of target periods

Although the database contains full length published records, allowing the investigation of transient climatic changes, our data synthesis in the frame of this manuscript is focused on key periods for which there is a specific interest in the paleoclimate modeling community: the last 200 years, the Mid-Holocene (MH; 6 ka), the Last Glacial Maximum (LGM) and the last interglacial period (hereafter LIG). The methodology used to estimate the isotopic offset between the different periods and the determination of its significance are provided in the appendix.

The last 214 years (1800 to 2013 CE, Common Era, noted as "last 200 years" for simplification) have been selected because (i) they encompass instrumental measurements (precipitation or seawater isotopic composition, air and water temperature, rainfall, sea level pressure…), and because (ii) isotopic atmospheric models can be nudged towards atmospheric historical reanalyses, thus providing a realistic framework for model-data comparisons (e.g. Yoshimura et al., 2008). It is here in fact extended back to 1800 to encompass, if possible, the climate response to the large 1809 and 1815 volcanic eruptions. This period is particularly important for detection and attribution of climate change, and, so far, the short duration of isotopic measurements in precipitation samples (i.e. at best 60 years for $\delta^{18}O$ in central Europe; Araguas-Araguas et al., 2000; GNIP Database, IAEA/WMO, 2015), has limited systematic investigation of recent trends. Here, we aim at expanding this documentation from highly-resolved proxy archives (mostly ice cores and tree-ring cellulose). Note that the records do not necessarily span the entire key periods (i.e. a record spanning only the last 50 years would be included in our statistics for the present-day period).

The MH (6 ± 0.5 ka, thousand years before 1950) has been selected as a target for
paleoclimate modeling (https://pmip3.lsce.ipsl.fr) as a compromise between the
magnitude of orbital forcing, and climate responses at the end of the glacial ice sheet
decay. The orbital configuration produces enhanced (reduced) insolation in the
northern (southern) hemisphere during boreal (austral) summer, associated with
warming in mid and high northern hemisphere latitudes as well as enhanced northern
hemisphere monsoons (Braconnot et al., 2012). So far, most quantitative model-data
comparisons for this period have focused on sea surface (Hessler et al., 2014) or
surface air temperature inferred from marine and pollen data, and precipitation
changes inferred from pollen or lake level data (Harrison et al., 2013). They suggest
that models tend to underestimate the magnitude of latitudinal temperature gradients,
as well as the magnitude of continental precipitation changes (Flato et al., 2013).
While the signal-to-noise ratio is often small, this recent period is well documented in
many well-dated, high-resolution archives, motivating a synthesis of proxy
information.
The LGM (19-23 ka) corresponds to a major global climate change, in response to
decreased greenhouse gas concentration and expanded continental ice sheets, with
an amplitude of global cooling of around 4°C, comparable to the magnitude of
projected 21[st] century high-end warming (Collins et al., 2013). Due to the magnitude
of the radiative perturbation associated with changes in atmospheric composition and
ice sheet albedo, this period is particularly relevant for climate sensitivity (Masson-
Delmotte et al., 2013; Rohling et al., 2012; Schmidt et al., 2014). Moreover, the LGM
has been widely investigated through well-preserved natural archives with improved
chronologies (Reimer et al., 2013). A synthesis of marine data has been achieved
within the MARGO collaborative effort (Waelbroeck et al., 2009), leading to a
database of multi-proxy sea surface temperature estimates, complementing surface
air temperature change between the LGM and present-day inferred from pollen and
ice core records (Braconnot et al., 2012). This period is marked by changes in the
thermohaline circulation (Duplessy et al., 1988; Shin et al., 2003; Yu et al., 1996),
large scale atmospheric circulation (Chylek et al., 2001; Justino and Peltier, 2005,
Murakami et al., 2008), El Niño - Southern Oscillation (ENSO; Tudhope et al., 2001;
Stott et al., 2002) as well as the monsoon and Inter-Tropical Convergence Zone
(ITCZ) position (Van Campo, 1986; Braconnot et al., 2000; Broccoli et al., 2006;
Leduc et al., 2009; Bolliet et al., 2011; Sylvestre, 2009). The large uncertainties
associated with changes in ocean circulation and their role for the carbon cycle and
the tropical water cycle have already motivated data syntheses and model-data
comparisons (Bouttes et al., 2012; Caley et al., 2014, Risi et al., 2010).
Finally, the last interglacial period (115-130 ka) is characterized by large changes in
orbital forcing, together with reduced volume of the polar ice sheets (Kukla et al.,
2002; Govin et al., 2012; Masson-Delmotte et al., 2013; Capron et al., 2014). While
global mean temperature is estimated to be less than 2°C warmer than today, based
on syntheses of temperature reconstructions and simulations (Otto-Bliesner et al.,
2013), northern hemisphere summer warming in this period can reach the same
magnitude of feedbacks than in future projections (Masson-Delmotte et al., 2011a). It
is also characterized by enhanced inter-hemispheric and seasonal contrasts
(Nikolova et al., 2013). Large uncertainties also reside on the conversion of
Greenland and Antarctic ice core water stable isotope records to temperature, with
implications for assessing the vulnerability of ice sheets to local warming (Masson-
Delmotte et al., 2011a; Sime et al., 2009 & 2013; NEEM community members, 2013).
Climate models have been shown to underestimate the magnitude of Arctic warming
and to fail capturing Antarctic temperature trends (Lunt et al., 2013; Bakker et al.,
2014). This may arise from vegetation and land ice feedbacks, which were not
resolved in the simulations. While all of the above motivate a proxy record synthesis
for this period, highly-resolved archives remain scarce (Pol et al., 2014), and large
age-scale uncertainties constitute a major obstacle, especially given the
asynchronous climate change detected in both hemispheres (Stocker, 1998; Masson-
Delmotte et al., 2010; Bazin et al., 2013; Capron et al., 2014).


**3. Database construction steps**
The first step consisted in gathering all the $\delta^{18}$O, $\delta^{13}$C and $\delta$D data available from the
two main online paleoclimate data depositories (NOAA and PANGAEA), together
with marine sediment records from the LSCE (Gif-sur-Yvette, France),
paleoceanography internal database (Caley et al., 2014) and literature survey and
personal communication (2013,2014) with authors. This work was performed from
May 2013 to July 2014.
A metafile has been built in order to list the main parameters of these datasets: core
name, reference, associated publication Digital Object Identifier (DOI), core site
latitude, longitude and elevation or depth coordinates. We have also inserted a flag to
describe the quality of age models for marine sediment cores (see next section). All
ages were converted into thousand years before present (ka), using 1950 CE as the
reference year. For each archive, we have stored the depth / age / proxy value data
into a separate three-column file. This protocol was applied to each archive and
proxy record. For instance, for a publication reporting $\delta^{18}$O time series based on four
different foraminiferal species, extracted from two deep sediment cores, we have
produced eight files, using a simple text tabulated standard format. This
standardization was adopted in order to facilitate the comparison of records, and to
allow future automated calculations. The name of this standard data file was inserted
into the metafile. The name of output files was established based on the name of the
original file provided by authors. We thus simply added the acronym "SIMPL" (for
"simplified") to the data-only file name. For publications presenting several records,
the different cores, species and/or proxies were indicated to the individual data files.
For       instance,       "*stott2007_MD81_cmund_corrected_SIMPL*"       and
"*Stott2007_MD81_cmund_SIMPL*" are the output files for the $\delta^{18}$O records from core
MD98-2181 published by Stott et al. (2007), based on the benthic foraminifera
*Cibicidoides mundulus* with and without adjustment for vital effect, respectively.
All the available information describing the associated age model was extracted and
compiled into a separate spreadsheet named after the original data file, with the
addition of the "TIEPTS" (for "tie points") to the file name, as well as the core
reference in case of articles based on multiple records. This spreadsheet contains
sample reference and depth, raw and/or calendar ages from radiometric dating with
the name of the species or the type of material measured, tie points used for core-to-
core correlation, and the amount of dated material. The name of this file was also
listed in the metafile, and this information was used to evaluate the age model (see
next section).
This database was used to calculate basic statistics (number of data points, average
proxy value, standard deviation) for the MH, the LGM, the last Interglacial, and for the
reference present-day climate (last 200 years).

**4. Age model evaluation**

**4.1 Deep sea sediment cores**
Following the protocol developed for the MARGO project (Waelbroeck et al., 2009),
quality flags were attributed to the chronology of the deep sea sediment cores and
speleothems. For this purpose, several factors were taken into account:
1. The density of chronologic markers: AMS [14]C and/or U-Th dates, core-to-core
correlation tie points, reference horizons (tephra, paleomagnetic excursions…).

2. The position of age markers, especially at the boundary of our target periods. For
instance, we consider that the LGM (19-23 ka) is better constrained with two AMS
[14]C dates at 19 and 23 ka than with four dates within the 20-22 ka interval.

3. The presence of sedimentary disturbances (turbidites, hiatus, bioturbation) and
post-deposition or coring events (gaps, core breaks, post-depositon reorganization of
speleothems crystals). This aspect of the age-model evaluation is however restricted
to the information provided by authors concerning the possible presence of such
disturbances.

4. The level of detailed description of the age model: raw [14]C and U-Th ages,
samples reference, type of material or species analyzed, reservoir age and
calibration program or curve used in case of marine material. Reservoir ages still
remain vigorously discussed (Soulet et al., 2011; Siani et al., 2013). Here, we used
the reservoir ages as originally published.

5. Marine Core-top constrains. It is customary among paleoceanographers to assign
"0 BP" to the uppermost sample of the core. Many late Holocene records are also
dated using extrapolated ages between the most recent datum and the top of the
core. This implies that the top of deep-sea cores is often poorly chronologically
constrained. Although arbitrarily dated, these data points were integrated to the
calculation of present-day average values.
6. For records older than the $^{14}$C reliability interval (~35 ka to 60 ka, where the
uncertainty on the calibration into calendar ages strongly increases, Plastino et al.,
2001; Bronk-Ramsey et al., 2013), the quality flags are based on the number of tie
points, and the type of material used for core-to-core correlation (e.g.: well dated high
resolution ice core vs. low resolution sediment core).
Quality flags ranging from 1 (very good) to 5 (poor) were therefore included in the
metafile for each deep-sea sediment core and speleothem dataset. This evaluation
protocol was not applied to archives such as tree rings, varved lacustrine cores, high
accumulation ice cores, modern corals or mollusk shells where annual counting
allows building accurate chronologies. We thus assigned the best quality flag to
these records.
In order to illustrate the chronological quality flag, we describe hereafter five
examples:
a) Quality flag 1 (excellent): Marine Core A7 (27.82°N, 126.98°E investigated by Sun
et al., 2005) is constrained by 15 well-distributed AMS $^{14}$C dates ranging from 1 to
17.5 ka, corresponding to the time period where oxygen stable isotope data are
available. There is therefore no significant arbitrary-dated interval. The authors used
a dated ash layer to establish a precise correction of the theoretical reservoir age,
and the effect of local turbidites was precisely monitored. The dating protocol is
described in detail, and reports samples labels, reservoir age, and the calibration
curve. Despite the lack of information on the selected species and the amount of
material used for $^{14}$C dating, we assigned the maximum quality flag to this age
model.
b) Quality Flag 2 (good): Marine Core GEOB3129/3911 (4.61°S, 36.64°W) is dated
through 16 AMS $^{14}$C dates spanning the 1.8-20 ka interval, which coincides with the
period covered by isotope measurements (Weldeab et al., 2006). The dating protocol
is relatively well described although reservoir ages and the amount of measured
material are not directly mentioned. With one date at 20 ka and another one at 16.9
ka, the distribution of dates does not provide a precise picture of the timing of the
starting date of the last deglaciation.
c) Quality Flag 3 (average): Marine Core KNR159-5-33GGC (27.56°S, 46.18°W;
Tessin and Lund, 2013) is constrained by 14 AMS $^{14}$C dates between 1.6 and 18.5
ka, and the entire dating protocol is well described. However, the AMS $^{14}$C dates are
not homogenously distributed, with only 4 data points within the 1.6-14 ka interval
and 10 dates between 15.4 and 18.5 ka. The chronology of the Holocene is therefore
poorly constrained. Moreover, anomalously old material is intercalated between
younger sediment, interpreted as deep burrying (Sortor and Lund, 2011).
d) Quality Flag 4 (below average): the age scale of Core RC10-196 (54.70°N,
177.08°E) is particularly well described by Kohfeld and Chase (2011). However, only
three AMS $^{14}$C dates and one $\delta^{18}$O data point for oxygen isotope stratigraphy  are
available between 10 and 22 ka, while the $\delta^{13}$C and $\delta^{18}$O records span a
considerably wider time interval (10-86 ka). The starting point of Termination I is not
well defined in $\delta^{18}$O, making the datum at 22 ka relatively imprecise. Although the
authors did not focus on the last deglaciation, we incorporated this record in the
database, because only very few records have been recovered in this part of the
North Pacific.
e) Quality Flag (poor): $\delta^{18}$O record from Core M44/3_KL83 (32.60°N, 34.13°E;
Sperling et al., 2003) spanning the last 13 kyrs. This record is constrained by only
one $^{14}$C AMS date (7.6 ka), leading to large uncertainties in the timing of the whole
Holocene.

**4.2 Other archives**
**Ice cores**
Dating ice cores is a crucial issue, as these highly-resolved archives are often
compared to marine cores and speleothems to assess the timing of climatic events
between high and lower latitudes. Ice core chronologies are regularly updated using
available age markers and dating is synchronized among different ice cores (e.g.
Rasmussen et al., 2006; Vinther et al., 2006; Ruth et al., 2007; Bazin et al., 2013;
Veres et al., 2013), with estimates of associated age scale uncertainties. For that
reason, it was decided not to attribute dating quality flags for ice cores chronologies
in this database. For the last interglacial period, LGM and MH, most ice cores
chronologies would be flagged as good to excellent, depending on the dating
strategy. For the last 200 years, the quality of ice core chronologies can vary from
excellent for high accumulation areas (where annual layer counting and volcanic
horizons are available) to good in the driest central Antarctic areas.

**Speleothems**

Dating speleothems generally involves radiometric methods or, in rare cases,
counting of annual laminae when they are visible. In the majority of cases, it is based
on uranium series methods (schematically $^{234}$U decays into thorium $^{230}$Th); when the
U/Th method is not possible because of too large detrital content, some authors may
use AMS $^{14}$C with a correction of dead carbon producing quite large errors. U-Th
method on speleothems can have a <1% 2-sigma error bar and the age limit of the
method is close to 450 ka; but depending on the detrital content of the calcite and on
the method used (i.e. TIMS, MC-ICPMS or alpha counting for old records), errors
may be variable. Chronologies based on radiometric dating were evaluated similarly
to what was done with marine cores, with quality flags based on the resolution and
distribution of the dated samples, and taking into account the possible sedimentary
issues (recrystallizations, hiatus not caused by climate fluctuations). In the case of
dating by lamina counting, similarly to what was done to modern coral records, we
considered that the error on the chronology is low, and assigned the maximum
quality flag to the age model of these cores.

**Lacustrine records**


The construction of age models for lacustrine cores is somewhat similar to what is
applied for marine cores. Most of the chronologies are based on AMS $^{14}$C dating
measured on carbonate or organic compounds. Similarly to what was performed for
marine datasets, the quality flags for lacustrine records are based on the density of
$^{14}$C dates and their position relatively to key transitions. We also took the
sedimentary disturbances (e.g. sedimentation hiatuses) into account as well as the
presence of potential corrections for residence time and reservoir effects revealing an
effort for considering the impact of the lake circulation dynamics in the sediment age.
The chronology of some of the compiled lacustrine records was performed by
counting of seasonal varve, generally resulting in a high accuracy (Sprowl, 1993). As
a result, we attributed the "excellent" quality flag to varve-based chronologies.

**Tree-ring records**

Tree-ring are generally short and well-dated records. The dating method is based on
precise counting of single rings produced each year by individual trees. Although
some chronologies can be affected by a few double or missing rings, tree-rings may
be the archive presenting the most robust chronologies and allow the attribution of a
precise calendar year to each of the rings. We therefore assigned the "excellent"
quality flag to all of the tree-ring records of our database.

**5. Interactive visualization tool**
NOAA and PANGEA open-access online libraries host a huge amount of
palaeoclimatic datasets, but browsing and downloading these data may sometimes
not be optimal. Each dataset must indeed be downloaded individually, without having
the possibility to quickly visualize the records online.

This is particularly critical when users need to download a large amount of records
not corresponding to a specific site and/or author. This lead us to develop a tool that
optimizes the datasets browsing step, with an online data plotting function, and a
user-friendly tool for downloading multiple datasets.

One of the main objectives of this application (http://climateproxiesfinder.ipsl.fr/) is to
ease exploration of multi-dimensional data assembled from mutiple proxy records
containing common features. This approach is relatively new and benefits from the
latest interactive data visualization techniques (d3.js [https://d3js.org/], dc.js
[https://dc-js.github.io/dc.js/], Leaflet [http://leafletjs.com/], bokeh
[http://bokeh.pydata.org]). Fig. 1a shows the layout of the Climate Proxies Finder
which consists of a world map (top row) and four charts representing, respectively,
the proxy depth, age (oldest, most recent), archive type (ice, lake, ocean,
speleothem, tree) and material  (e.g. carbonate, coral, etc.). A table of the available
records is also displayed at the bottom of the screen (first 100 only). This table
displays information about the records (depth, age [most recent, oldest], archive,
material, DOI, and the reference of the corresponding scientific paper). The DOI is
hyper-linked to the google scholar search engine. The user can interactively filter the
dataset by clicking or brushing on any of these charts or by dragging and zooming in
and out of the map. Since all charts are inter-connected, they will automatically be
updated according to the filter selections. Fig 1b shows an example of this interactive
filtering with the selection of ocean archive type near the surface (0 - 500 m).
Accordingly, due to the crossfiltering functionality, all other charts and the table reflect
only the proxies selected by these filters. This application also allows the user to
display an interactive plot of the time series of the available isotopes by clicking on a
map marker (see Fig. 1c for an illustration).
Lastly, the user is able to download in a zip file the selected proxy data as CSV files
and time series plots by clicking on the shopping cart icon.
The Climate Proxies Finder application continues to evolve as new features are
needed, such as adding a filter for proxy chronological information quality.
**6. Results**
The overall increase in the number of records and publications per year over the last
50 years (Fig. 2) reflects the growing investment in obtaining stable isotopes records
to document and understand past climates. The peak in the number of records
published in 1998 and 1994 are mostly due to the presence of some publications
compiling a large number of previously unpublished marine records from the Atlantic
Ocean (Sarnthein et al., 1998; Sarnthein et al., 1994).

## 6.1 Geographical distribution of data and temporal resolution

This section briefly describes the status of the database for marine and terrestrial records (Fig. 3), and provides a synthesis of stable isotope data for each focus period.

A total of ~6,400 records were collected from the NOAA and PANGAEA data repositories as well as from the internal LSCE database. About 3300 marine records were rejected, as they are not yet published. Following the settings of our online portal, we also isolated about 300 $\delta^{18}$O and $\delta^{13}$C published records not dated (~200 records) or containing no information about the core site elevation or depth (~100 records). We thus accumulated about 1,700 $\delta^{18}$O records from ~900 sites, about 900 $\delta^{13}$C records from 450 sites, and about 230 $\delta$D records from 60 core sites (with 20 additional deuterium excess records). When considering the different types of archives, we compiled about 1,200 $\delta^{18}$O and ~700 $\delta^{13}$C records from 600 marine sediment cores, 200 $\delta^{18}$O and 75 $\delta^{13}$C speleothems records from 60 caves, 200 dated $\delta^{18}$O records from 50 ice cores (with about 60 additional dated $\delta$D datasets and ~20 $\delta^{17}$O records), 60 $\delta^{18}$O and 60 $\delta^{13}$C lacustrine records (with $\delta$D datasets), as well as 85 $\delta^{18}$O and 80 $\delta^{13}$C records from tree rings.

Among all the 1,900 collected marine records, about 850 do not present any information about the construction of their age model and about 950 records are associated with age model tie points or by default associated with an *excellent* chronology (e.g. modern corals), while most of the lacustrine cores and speleothems are associated to chronological information. We also note that, when not considering tree-rings records, about 500 dated records do not present any sampling depth or distance scale. The absence of the age scale and/or chronological tie-points clearly prevents any comparison with other records or with climate model output. Similarly, the absence of a depth scale prevents the detection of potential sedimentary or chronological issues, and therefore the correction with existing age models.

### 6.1.1 Geographical distribution


For each period of interest, although the amount of compiled records is large enough,
the geographic distribution of marine cores is not homogenous, as 75% of the $\delta^{18}$O
and $\delta^{13}$C dated records are located in the Northern Hemisphere, with a maximum
density in the northern sub-tropical band (Fig. 3 and 4). The Atlantic Ocean is the
best documented (about half of all marine records). Most of the compiled records for
the Indian, Pacific and Southern Oceans come from sediment cores recovered on
continental margins, because a part of the seafloor in the open ocean is deeper than
the carbonate compensation depth in these basins (Berger and Winterer, 1974), and
the sedimentation rate is particularly low in the large oligotrophic areas of the open
ocean. This lack of suitable core sites constitutes a critical limitation for the
documentation of the past open-ocean circulation and mechanisms affecting the
entire Indian and Pacific basins, such as ENSO, latitudinal migrations of the ITCZ,
fluctuations in the thermohaline circulation, with possible formation of past North
Pacific intermediate and deep water (Mix et al., 1999; Ahagon et al., 2003; Max et al.,
2014), and storage of carbon in the Southern Ocean (Skinner et al., 2010; Burke and
Robinson, 2011). Vast areas remain virtually undocumented in the Indian, Pacific and
Southern Oceans. A large majority (about 90%) of the records of the ocean database
are based on foraminifera, while corals are much scarcer and only few studies use
molluscs or diatoms.

The distribution of continental records (Fig. 3) naturally depends on the position of
caves, lakes, forests as well as ice sheets and glaciers. Speleothem $\delta^{18}$O records are
found on each continent, but with a very heterogeneous distribution. In fact, due to
the distribution of caves presenting exploitable speleothems, several large areas
(Russia and central Asia, northern and tropical Africa, Canada, central South
America) remain undocumented, while the density of records is large in Europe,
USA, Central America and China. While they have provided highly resolved records
of regional climate variability (e.g. the monsoon and ITCZ, circum-mediterranean
continental climate), speleothems do not provide a global coverage. Lacustrine
records are also very unevenly distributed, with very few dated isotopic records in
South America, Africa, Russia and Australia, although these regions present
numerous lakes.
Oxygen and carbon stable isotopes from tree rings cellulose have recently emerged
as powerful paleoclimate proxies, albeit with heavy sample preparation (Libby et al.,
1976; Long, 1982; Ehleringer and Vogel, 1993; Switsur and Waterhouse, 1998). This
feature, and the fact that few tree ring isotopes datasets are available online, lead to
relatively scarce archives at a global scale. Most of the available records are located
in Europe, while the remaining other datasets (mainly $\delta^{13}C$ records) are restrained to
a few sites in Asia, South America, Siberia, Costa Rica and USA. This distribution of
records implies that associated large-scale climate reconstructions are somewhat
constrained to Europe.
With respect to ice cores, 75% of the compiled $\delta^{18}O$ and $\delta D$ are from Greenland and
Antarctica. Few cores indeed were recovered from high elevation ice caps and
glaciers from the Andes, Alaska, Arctic Russia, Svalbard, Mount Kilimanjaro and the
high-latitude Canadian islands, close to Greenland (Fig. 3). We stress the fact that
most published ice core records from Tibet spanning the past centuries are not
available from open-access sources.

Contrary to the geographical distribution, the vertical distribution of marine cores
along the water column is relatively homogenous for the global ocean (Fig. 5), with
more than 100 datasets in each of the 500 m-thick layers from the surface down to
4000m, while data are scarce below this level.


**6.1.2 Temporal distribution**

We now describe the distribution of records throughout the different periods of
interest (Fig. 6). Marine $\delta^{18}O$ and $\delta^{13}C$ records are well represented over the four
periods, with at least 200 records available for each of the time slices. However,
many marine sediment core tops are poorly dated, and thus the number of marine
data delivering a robust characterization of recent oxygen and carbon isotopic
composition is limited. About half of the marine records have only one data point over
the last 200 years (about 50% of the $\delta^{18}$O records and 60% of the $\delta^{13}$C records) and
most of them have fewer than ten data points over the last 200 years (~65% of $\delta^{18}$O
and $\delta^{13}$C records). When considering the other PMIP key periods, it appears that the
distribution is similar for the MH (about 90% of the $\delta^{18}$O and $\delta^{13}$C records have fewer
than ten data points), while the resolution is slightly better for the LGM (65% of $\delta^{18}$O
and 70% of $\delta^{13}$C records have fewer than ten data points) and for the large time
interval assimilated here to the last interglacial (~50% records have fewer than ten
data points).
Speleothem records span a large variety of time-intervals, ranging from seasonal to
glacial/interglacial scale. Due to the heterogeneity of the time slices spanned by
speleothems records, the information provided is relatively fragmented. As a result,
although we compiled more than 200 speleothem $\delta^{18}$O records, none of the four key
time-slices selected by the PMIP project contains more than 60 records (30 for $\delta^{13}$C),
due to the fact that many records span time intervals are in between these time-
slices. Also, only three dated speleothem $\delta^{18}$O records span the entire time interval
from the last interglacial period to present-day, and only 14 records span both the
LGM and the MH. In general, speleothem records have a better temporal resolution
than marine records. For each of the four key periods, at least 60% of the records
display more than ten data points. One difficulty arises from the fact that
exceptionally long speleothem records such as the one obtained from the Hulu and
Dongge caves records (Wang et al., 2001; Wang et al., 2005) have been obtained
from the compilation of measurements performed on several speleothems/cores from
one single cave. These multiple individual cores may present significant and varying
offsets which can be identified over different periods of overlap (see Wang et al.,
2001 and Yuan et al., 2004). As a result, establishing a robust composite record
allowing calculation of anomalies between different past periods is particularly
delicate for these archives. For this reason, we decided to keep the individual short
datasets separated as they were published, and did not build long and continuous
composite records. Therefore, composite records cannot be displayed in our LIG-MH
comparison map (Figure 9).
$\delta^{18}$O records from ice cores are relatively scarce for the oldest PMIP time slices, with
only ~45 records spanning the MH, ~40 for the LGM, and 14 concerning the LIG (13,
13 and 6 for $\delta$D, respectively). Only five $\delta^{18}$O records are continuous from the LIG to
the Holocene. Ice core records however provide a wealth of information on the spatial
and temporal variability of surface snow isotopic composition over the last decades,
as about 140 of the ~180 compiled $\delta^{18}$O dated records spanning the last 200 years
exhibit at least ten data points within this period (50 out of 55 dated records for $\delta$D
and deuterium excess).
As the effect of burial on $\delta^{18}$O of fossil wood cellulose remains poorly known (Richter
et al., 2008), we selected records exclusively based on living trees or timber wood.
Consequently, the compiled records from tree ring cellulose can only be used to
monitor the climate fluctuations of the last millennium at the very best. We have
identified ~80 tree ring cellulose $\delta^{18}$O records which cover the past 200 years (~80
for $\delta^{13}$C). Most of the records have been provided at seasonal to decadal temporal
resolution.
Lacustrine cores are generally short and records generally span relatively limited time
intervals. As a result, only the PD and MH are covered by a relatively large number of
records (~35 $\delta^{18}$O, 30 $\delta^{13}$C and ~135 $\delta$D records for PD; 25 $\delta^{18}$O, 30 $\delta^{13}$C and ~45
$\delta$D records for MH), while datasets spanning the LGM and LIG are very scarce (30
when considering $\delta^{18}$O, $\delta^{13}$C and $\delta$D records).

**Datasets temporal resolution**

Supplementary Fig. A1 (see Appendix) shows the variety of temporal resolutions in
the compiled records spanning the past 200 years (1800-2013 CE). Dating of marine
sediment core tops remains a critical issue, due to alterations during the coring
process as well as sediment reworking and bioturbation. In fact, the upper first
centimetres are generally water-soaked and thus often lost or altered during the
recovering of marine cores, which, in case of moderate or low sedimentation rates,
leads to the loss of material spanning the last hundreds or thousands years.
Additionally, bioturbation can alter the upper sediment down to 10 cm below the
water-sediment interface (Boudreau, 1998). As a result, many core tops provided as
present day references might actually reflect older conditions (from several centuries
to few millennia, Barker et al., 2007; Löwemark et al., 2008; Fallet et al., 2012).
Solving these issues might require a precise investigation of bioturbation tracks in the
upper layers of sediment cores and drastic improvement of the coring and analysis
techniques, as suggested by the final conclusions of Keigwin and Guilderson (2009) :
"Until we can directly radiocarbon date individual foraminifera, the role of bioturbation
will always be a problem in core top calibration studies". These sedimentary issues
are often accompanied by insufficient resolution and quality of the sediment core-tops
dating procedure. In fact, present-day conditions are represented by only one data
point in about half of the datasets, generally dated via linear extrapolation of deeper
tie-points. About 95 marine $\delta^{18}O$ and 35 $\delta^{13}C$ records exhibit a decadal to annual
resolution, generally arising from corals (65% of the records) with robust layer-
counted annual chronology.
While chronology is not an issue for tree ring cellulose records, the number of
individual tree samples combined for each year can be a limiting factor. Several
studies have investigated the signal to noise ratio, and demonstrated the importance
of combining at least 4-5 trees from a forest to extract the common climate signal
(e.g. McCarroll and Loader, 2004; Daux et al., 2011; Labuhn et al., 2014). The same
issue arises for ice core records, especially for the past centuries when the noise
caused by processes such as wind scouring can be significant when compared to the
small climatic signal (e.g. Fisher et al, 1985; Masson-Delmotte et al., 2015). As a
result, the records resulting from stacks combining several ice cores from a given site
have stronger climatic relevance than records based on individual ice cores.
However, the non-polar ice cores experience their best dating on this period. The
dating is usually based on the multi-proxy annual layer counting which is based on
the seasonal variations of insoluble particles and the isotopic composition of ice.
Moreover, the natural radioactive material decay of suitable radionuclides ($Pb^{210}$ for
example) and the identification of prominent horizons of known age from radioactive
fallout after atmospheric thermonuclear test bombs ($Cs^{137}$, $Sr^{90}$, $Am^{241}$) provide
absolute reference horizons, and are currently used in the Southern Hemisphere
(Vimeux et al., 2008, 2009a for example in the Andes).
Several recent speleothem and short ice core records benefit from annual layer
counting, with an accurate chronology, but this is not systematic. Ice core datasets
encompass a large proportion (~70%; 120 records) of highly-resolved (decadal to
annual) records, while this percentage is significantly reduced for speleothems (about
one half of the 90 records spanning the last 200 years).

For the MH and LGM, marine records also have the lowest temporal resolution, as 80% of these datasets exhibit 4 data points or less over the 5.5-6.5 ka interval, and none of the records are available with a resolution better than respectively 20 and 40 years (Fig. A2 and A3 in Appendix). Ice core records spanning the MH and the LGM are relatively scarce (55 and ~50 datasets, respectively), and most of them exhibit decadal to centennial resolution. Speleothem records are slightly more abundant than ice core records (90 and 55 records for the MH and the LGM, respectively), with very variable resolution, from millennial to sub-decadal. Speleothems and ice core records spanning the Last Interglacial are scarce (about 35 and 15 records, respectively; Fig. A4 in Appendix) and only some of them present a centennial resolution or better, while marine records are abundant, but most of them have millennial or lower temporal resolution.

Lacustrine data can roughly be divided into two groups, with about half of the records covering only the last decades, while the other records are generally much longer, spanning the Holocene period, and few datasets cover the glacial period.

The present day is somewhat well resolved, as about 65 % of the $\delta^{18}$O and $\delta^{13}$C records spanning this time interval exhibit at least ten data points. This trend is also observed for the MH, with about 65 % of the records presenting ten or more data points. $\delta$D records appear to be much less well resolved, mostly because a large number of records originate from surface sediment studies based on dated core tops, resulting in a single data point. As a result, only 20 % on the $\delta$D records show at least ten data points for the PD. This lower resolution for $\delta$D is also verified for the MH, as none of the records present more than ten data points.

**Age model quality evaluation**

Results from the evaluation of the quality of chronologies are highly variable from marine and lacustrine cores to speleothems (Fig. 7). The overall quality of age models for marine records is moderate. In fact, we note that most of the records published in the 20[th] century present a missing or crude age model based on an insufficient number of AMS [14]C dates, with a lack of reported technical information. Although this result is somewhat deceiving, the quality of age controls has strongly improved during the last 15 years, thanks to better dating technologies and the

growing awareness of the absolute necessity to publish robust and well detailed
chronologies to precisely reconstruct past climate fluctuations.
Age models in speleothems are much better constrained, as most of the records
present an "excellent" or "good" quality flag. Speleothem records are indeed
generally constrained by abundant U-Th dates and authors often provide highly
detailed technical information. Age anomalies such as age reversals, outliers and
hiatuses are nevertheless identified in many records. These anomalies can be
caused by analytical issues (e.g. sample contamination, Th adsorption; Musgrove et
al., 2001; Wainer et al., 2011) or natural factors occurring simultaneously or after
sedimentation process (diagenetic alteration). Hiatuses may be induced by climatic
(e.g. severe droughts or permafrost impacts) or post-deposition (e.g. carbonate
dissolution) factors (Lachniet, 2009; Breitenbach et al., 2012).
The age models of lacustrine records are relatively good overall, with however larges
discrepancies in the quality of chronologies, depending on the dating technique. In
fact, some lacustrine records are dated by counting annual/seasonal varves or
laminations, leading to an excellent chronology. This dating technique is however
generally limited to relatively short records. Records providing longer signals (i.e.
spanning several thousand years) are generally dated by AMS $^{14}$C dates. Similarly to
what is observed for marine core dating, we note the possible lack of technical
information in publications, as well as limited resolution of dates, which prevent the
establishment of robust age-models. Also, the potential adjustment applied to $^{14}$C
ages to correct from radiocarbon reservoir and residence time effects is not
systematically provided, as well as the presence of possible hiatuses.

## 6.2 Changes between PMIP key periods

$\delta^{18}$O from oceans and atmospheric water (and therefore continental archives) vary in an opposite directions with climate fluctuations. We thus reversed $\delta^{18}$O fluctuations from ocean records in order to map coherent $\delta^{18}$O trends from all the different archives. We however report the original values in the text. We report anomalies with respect to the MH for coherency.

### 6.2.1 Changes between MH and Present-Day

The relatively large number of dated $\delta^{18}$O datasets covering both the last 200 years (PD) and the Mid-Holocene (MH) allows us to estimate possible offsets between these two periods (MH-PD; ~100 records from 70 sites; Fig. 8). We restrict the record selection to datasets presenting multiple data points for each of the two periods of interest, thus documenting both the signal (average value) and noise (standard deviation). Results indicate a large dispersion of data, ranging from large positive to negative offsets, while most of the records depict in fact very similar values for the two periods. This feature reflects the spatial heterogeneity of the response to climate changes, making particularly difficult the establishment of large-scale patterns. In a given region, differences also emerge between records from different archives (e.g. opposite sign of changes in speleothem vs lake records in Eastern Europe). The average difference is low in ice cores, but the overall negative offset observed in ice cores indicates a polar cooling during the last 6 ka, except around the Ross Sea in Antarctica. Particularly remarkable is also the positive anomaly from Chinese speleothems, commonly attributed to changes in Asian summer monsoon with a decrease in rainfall amount through the Holocene (Cai et al., 2010). The standard deviation of the data for the two periods of interest are however quite large in most cases. In fact, in the three types of archives, this noise is either of the same order or higher than the calculated PD-MH offset. As a result, the relatively weak isotopic change between these two periods is not significant in 2/3 of the records. Because we did no account the analytical error associated with $\delta^{18}$O measurements (as this indication was missing in some of the datasets), we may underestimate the noise level, and thus the number of records presenting an insignificant PD-MH offset.

### 6.2.2 Changes between the Last interglacial and MH

We now apply the same approach for the change between LIG and MH (Fig. 9). This relies on 75 $\delta^{18}O$ records from ~45 sites presenting multiple data points for both of the two periods of interest. We observe more enriched continental (more depleted marine) $\delta^{18}O$ values for LIG than during the MH in ~20 records, suggesting relatively warmer conditions during LIG, with no apparent geographical trend. However, about half of the LIG-MH anomalies are in the range of the natural standard deviation, and thus cannot be considered as statistically significant. Considering only the records presenting a significant offset nevertheless suggests warmer conditions (enriched continental and depleted marine $\delta^{18}O$) values during the LIG than MH.

Recent syntheses have shown contrasting results in temperature changes between the Last Interglacial period and present day (e.g. Otto-Bliesner et al., 2013), with positive temperature anomalies at both poles, but not occurring simultaneously (Capron et al., 2014), and negative temperature anomalies in some tropical areas. Contrasted regional patterns are expected from the different orbital configurations. Several studies have also highlighted a large magnitude of climate variability during the LIG period (Cheddadi et al., 1998; Lototskaya and Ganssen, 1999; Hearty et al., 2007; Rohling et al., 2007; Pol et al, 2014).

### 6.2.2 Changes between the LGM and MH

Due to the limited amount of well-dated marine $\delta^{18}O$ records covering both the LGM and present day with more than one data point, we compare the LGM and the Mid-Holocene for investigating the isotopic amplitude of last termination (Fig. 10). The LGM-MH comparison reveals a significant negative (positive) offset in almost all the terrestrial (marine) records, with only few speleothem and coral records showing the opposite trend, mostly in the subtropics where they may reflect precipitation or atmospheric circulation effects rather than local temperature variations.

The highest deglacial amplitude is recorded in high elevation and polar ice core records, while the offset is less marked in oceans and speleothems. Marine datasets reveal a latitude-independent general amplitude of ~1.45 ‰ (1.55 ‰ when

considering only foraminiferal records, with a similar average value for benthics and planktonics), out of which ~1 ‰ is due to the change in land ice volume. In addition, we observe specific regional patterns. Larger amplitudes are identified in marine records from the north and South-East Atlantic (about 1.7 ‰), which contrast with smaller amplitudes in the tropics (~1.5 ‰) and maximum signals in the Mediterranean Sea (about 2.5 ‰). In this basin, this strong isotopic change is understood to reflect large SSTs deglacial warming and salinity changes induced by shifts in the regional atmospheric circulation (Bigg, 1994; Emeis et al., 2000; Hayes et al., 2005; Mikolajewicz, 2011). Statistics based on benthic foraminiferal $\delta^{18}O$ records (including datasets presenting only one data point in the periods of interest) reveal that there is no influence of core site depth on the amplitude of the LGM to MH transition ($R^2$ = 0.0029; n= 180).

Ice cores records from high latitudes are all marked by a -3.3 to -7.7 ‰ $\delta^{18}O$ shift, with however regional differences such as East-West gradients in both Greenland and Antarctica. Such regional differences may be induced by changes in ice sheet topography and different amplitudes of surface elevation changes at different locations (e.g. Vinther et al., 2009). Similar mechanisms may be at play in Antarctica, but remain poorly documented (e.g. Masson-Delmotte et al, 2011b). There is also evidence for regional differences in the response of Antarctic temperature to climatic changes (Turner et al., 2005, Steig et al., 2009; Steig and Orsi, 2013). The larger amplitude of glacial-interglacial isotopic changes in West Antarctica has been suggested to reflect regional processes coupling the Southern Ocean, sea ice extent and atmospheric heat transport (WAIS Divide Project Members, 2013). It is worth noting that Andean ice cores spanning the last glacial-interglacial transition show a similar deglacial isotopic shift (Vimeux, 2009b). The water stable isotopic composition in those ice cores is likely reflecting precipitation changes at regional scale and such a similar deglacial structure is explained by simultaneous cold conditions in the high latitudes and wetter conditions in the Andes (Vimeux et al., 2005; Chiang and Koutavas, 2008).

Different patterns emerge from speleothem records covering the LGM and MH, as only half of the datasets are marked by a more depleted glacial $\delta^{18}O$ level. Depending on the location, speleothem calcite $\delta^{18}O$ may reflect either paleotemperature and/or past changes in atmospheric water cycle (including

precipitation and circulation). Additional site-specific factors (cave microclimate,
mixing and evaporation of source waters through the soil and the epikarst, kinetic
fractionation during carbonate precipitation) may also influence the signal (Lachniet,
2009). Regional effects may also be at play in the western Middle East, where
speleothem records can be directly influenced by changes in the Mediterranean or
the Black Sea, which had diverging oceanographic evolutions between the LGM and
the MH, with the opening of the Bosphorus Strait. Individual records must therefore
be understood in their own regional environmental context, a feature also evidenced
by different amplitudes of change arising from different source archives. Thus, Fig. 8-
10 might be considered as an inventory of the available datasets, rather than a
cartography of the amplitude of climatically-relevant signals, expected to be
representative of the amplitude of annual mean precipitation or sea water isotopic
composition changes.

**7. Conclusions, recommendations and perspectives**

Our compilation of hundreds of records from different sources highlights the needs
for a standardized protocol of data storage. The output files provided by the different
depositories have different archiving formats. Several ongoing projects rely on
massive and automated extraction of datasets provided by authors. This effort would
be made easier if the data and publication information (core site specifications,
references, article title and abstract) were stored in individual CSV (Comma-
Separated Values) text files, rather than within files specifically designed for
spreadsheet software (e.g. Microsoft Excel/Apache OpenOffice), sometimes
containing several spreadsheets, that may not be readable by automated data
extraction programs. We also think that building a fixed disposition for datasets
constitutes a preliminary step and that it is essential for the existing and future data
depositories to find an agreement for an harmonized disposition, structure and
labelling for metadata and age modelling data storage. Some projects are following a
promising philosophy of homogenously-structured metadata (e.g. LiPD; McKay and
Emile-Geay, 2015). We highly encourage these constructive initiatives, as it becomes
urgent for the palaeoclimate research community to definitively adopt a universal file
format and metadata disposition, and define the type of contents to be included,
before starting compiling data, otherwise this will lead to a high risk of incompatibility
or of conflicting information from different sources or projects. Adopting this universal
format will however necessitate a clear agreement between data producers, users,
and compilers, as it requests at the end a unique structure compatible with all types
of archive and proxies, which may lead to some complications due to the variable
number of parameters to be included for each proxy and archive. When a universal
standard format will be definitively adopted, the conversion of our metadata
spreadsheet into a hierarchical structured may be relatively easy and fast.

Divergences in data units also constitute a major obstacle for automated extraction,
inter-comparison of records, and model-data comparisons. An illustrative example is
the use of various time units (years CE, years or before 2000 CE, years before 1950
CE, kiloyears BP, and million years BP). The establishment of standard time units for
palaeoclimatology such as use of ka (calendar kiloyears before 1950) would avoid
errors and homogenization of future datasets. Several discrepancies also exist with
respect to the geographical coordinates of core sites. Although the most common
format found in the literature is DMS (Degrees, Minutes, Seconds; e.g.: 25°22′34″N,
38°16′43″W), it is not supported by most mapping programs. Here, we converted all
the geographical coordinates into decimal degrees. We again highly encourage the
adoption of a standard notation, with the systematic presence of the decimal degree
version of the coordinates; we observe that an increasing number of authors now
provide both DMS and decimal formats.

Gathering information about the age models was a particularly critical step of the
construction of this database, in particular for the inclusion of lacustrine and deep sea
cores as well as speleothems. We highly encourage the authors to systematically
provide both depth and age scales as well as a comprehensive description of the
methodology used to establish the age scale, when available. While our earlier
comment was centered on deep sea cores, the same features apply for the
description of lake sediment cores, ice cores and speleothem chronologies. Even if
the methodology developed for the successive chronologies of deep ice cores is
usually precisely documented, no standardized reporting protocol exists for ice cores
from tropical and temperate glaciers. There is however no existing standard
procedure for the description of age models. The available information is often
fragmented, with missing information (raw AMS $^{14}$C dates, calibration program/curve
used to compute calendar ages, species used for analysis, amount of material
measured, marine reservoir ages, tie points, identification of hiatuses in
speleothems...). A standardized format including all the information related to the
establishment of the age models would be a major step forward. Finding a common
structure might however constitute a fastidious task, particularly because the samples
dating techniques are radically different for the different types of archives. A first  step
would constitute in finding a standard structure to be adopted for AMS $^{14}$C
measurements performed, for instance, on speleothems, marine and lacustrine
cores. Many old records are associated to very limited information concerning their
chronology, which prevents any tentative to reproduce the age model. Consequently,
it becomes necessary to adopt a common format which would be interoperable
between the different data repositories, and would include all the necessary
information to recalculate age models. For age models based on AMS $^{14}$C dating, we
suggest that the following information should become mandatory:

- Core ID
- Sample ID, lab name
- Sample depth with indication of any depth correction
- Type of material analysed, including species.
- Indication of sedimentary disturbances (hiatuses, turbidites, tephras, etc…)
914        and their corresponding depth
- AMS $^{14}$C ages and the associated error
- Calibrated ages and the associated error
- Program/calibration curve used for $^{14}$C dates calibration
- Reservoir age for marine cores, and the associated uncertainties
- Dates removed from the construction of the age model and the reason why
920        they were eliminated.


Additional information might include the type of equipment used for analysis and the
date of measurement, the posterior probability distributions of $^{14}$C dates, the
treatment applied for sample cleaning and the amount (weight or number of
specimens) of material analysed.

We have noticed a clear improvement of the quality of age models and of dating
techniques description during the last two decades, and most of the low quality
chronologies were published more than 20 years ago. This improvement of age
models is particularly critical with respect to the sequences of events during fast
transient climate reorganizations. In fact, previous studies have shown that many
past major climate changes involved abrupt responses (e.g. de Menocal et al., 2000;
Genty et al., 2006; Carlson et al., 2007; Zuraida et al., 2009; Clark et al., 2012; Rach
et al., 2014) as well as short delays between different proxy records and regions, like
the vigorously debated date and triggering of the onset of Termination I (Schaeffer et
al., 2006; Stott et al., 2007; Koutavas and Sachs, 2008; Smith et al., 2008; Bromley
et al., 2009; Clark et al., 2009; Shakun et al., 2012; Parrenin et al., 2013). In this
context of successive rapid climatic events and keeping in mind the growing interest
on transient climate simulations, it thus becomes necessary to have a large amount
of precisely dated and well defined records. Reservoir ages remain a critical issue in
palaeoceanography as well as their uncertainties. Many efforts have been deployed
during the last decade to better estimate reservoir ages. Several publications have
also suggested changes in reservoir ages between glacial and interglacial periods
(e.g. Waelbroeck et al., 2001; Bondevik et al., 2006; Sikes et al., 2016). In this
context, the age model of many old records may be outdated, and even considered
to be wrong. Unfortunately, the lack of information concerning the construction of
these initial age models makes the construction of an updated age model virtually
impossible. In this study, we did not aim to evaluate the accuracy of published
reservoir ages, which remain sometimes vigorously debated. We encourage authors
of publications to systematically justify their choice of a reservoir age, to describe the
associated uncertainties, together with the detailed age model information.
Our database may in the future allow the implementation of statistical age models
built on the existing age markers. Reporting the exact number of source records for
tree rings and ice cores is also important with respect to the signal to noise issue; this
is not always a standard practice.

Our software tool was designed to make the update of the database user-friendly and
easy, in order to allow future extension. Indeed, major synthesis efforts as the
MARGO project (Waelbroeck et al., 2009) are time limited (MARGO only includes
records published prior to 2005). Options for an automatic update include a regular
browsing of new published data, but we highly encourage authors to upload their new
data in our database using the user-friendly interface on the online platform. This
constitutes a fast and easy way to disseminate new data and increase their visibility,
and a unique opportunity for the scientific community to access and exploit newly
published datasets. This allows "data producers" to easily compare their records with
other existing records in a given area or at the global scale, and climate modelers to
access easily the data, and to the source references and their authors.

In the future, and if manpower resources are available, the database and web
interface could be easily opened to other proxies (paleotemperature proxies and
nitrogen isotopes for seawater, $CO_2$ and $CH_4$ from ice cores, tree rings width and
boreholes, pollens, circulation tracers such as $^{14}C$ and Pa/Th, etc.) of past and future
datasets. We also hope that our database, associated with current and upcoming
projects focusing on time-series age control (INTIMATE PROJECT, COST Action
ES0907) and chronological data managing (Mulitza and Paul, in prep.), would in the
future facilitate the use of paleoclimate datasets for data comparison and integration
into models with an homogenous and robust chronological frame. This is expected to
strengthen the use of proxy information for model-data comparisons, a topic
promoted in the Stable Water Isotope Intercomparison group (SWING) and the
isotope modeling working group of the Paleoclimate Modelling Intercomparison
project, with the potential to better document projections (Schmidt et al, 2014).

**Acknowledgements**
This study was supported by a national grant from the Agence Nationale de la
Recherche under the "Programme d'Investissements d'Avenir" (Grant #ANR-10-
LABX-0018) within the framework of LABEX L-IPSL.

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

 **Figure captions**

 **Figure 1**: Web portal screen captures illustrating the search criteria (A), the resulting

 maps (B), and the time series plot (C).

 **Figure 2**: Number of publications and records in the database versus year of

 publication.

 **Figure 3:** Map indicating the position of archives with different symbols representing

 the type of archive for dated $\delta^{18}O$ (top) $\delta^{13}C$ (center) and $\delta D$ records (bottom)

 available on the online portal. Note that these maps only display the location of dated

 records, and stack and multi-sites composite records are not included.

 **Figure 4**: Diagram showing the distribution of ice cores, tree-ring, lacustrine,

 speleothem and marine records as a function of latitude (°).

 **Figure 5**: Diagram showing the distribution of ice cores, tree-ring, lacustrine,

 speleothem and marine records as a function of coring site elevation.

 **Figure 6**: Diagrams showing the number of $\delta^{18}O$ and $\delta^{13}C$ records from marine and

 lake cores, speleothems, ice cores, and tree ring cellulose for each PMIP time slice.

 **Figure 7**: Location of lacustrine (triangles), speleothems (squares) and marine

 records (circles) where chronological information is available, and with quality flags

 for age model quality evaluation.

 **Figure 8**: Map showing the location of $\delta^{18}O$ records spanning the MH and PD with

 the symbols reflecting the type of source archive and colors documenting the

 amplitude of $\delta^{18}O$ variations between these two periods (MH-PD). The bottom figure

 shows the color scale as well as the fraction of records as a function of the MH-PD

 $\delta^{18}O$ anomaly. Note the non-linear scale for $\delta^{18}O$ difference. The $\delta^{18}O$ difference from

 marine records was reversed for coherency with the sign of changes of terrestrial

records. Note that some proximate core-sites may not be visible on the figure
because of graphical overlaps.

**Figure 9**: Same as Fig. 8 but for the difference between LIG and MH values (LIG-
MH).

**Figure 10** : Same as Fig. 8 but between the MH and the LGM (LGM-MH).

**Figures**

**Figure 1**

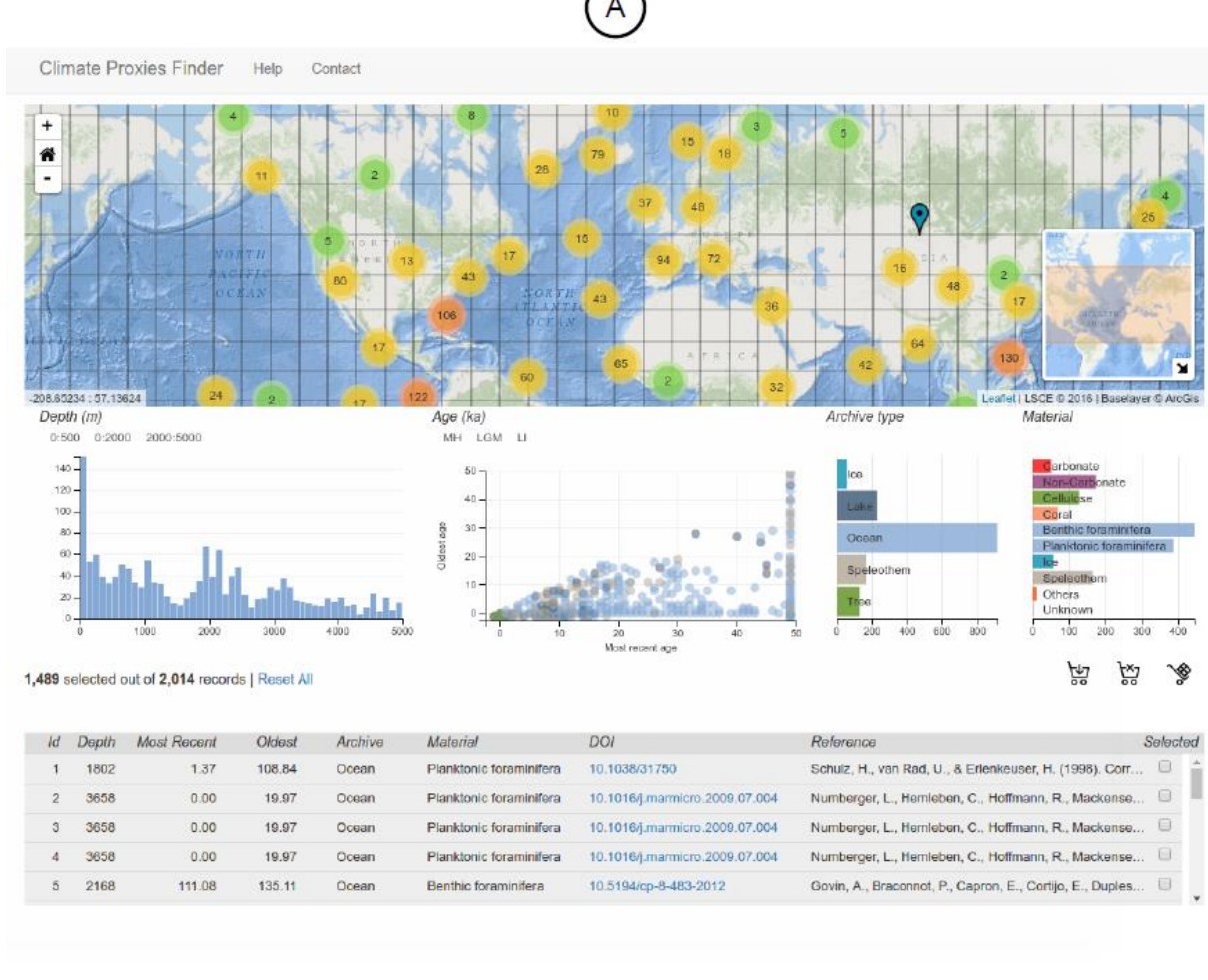


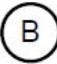

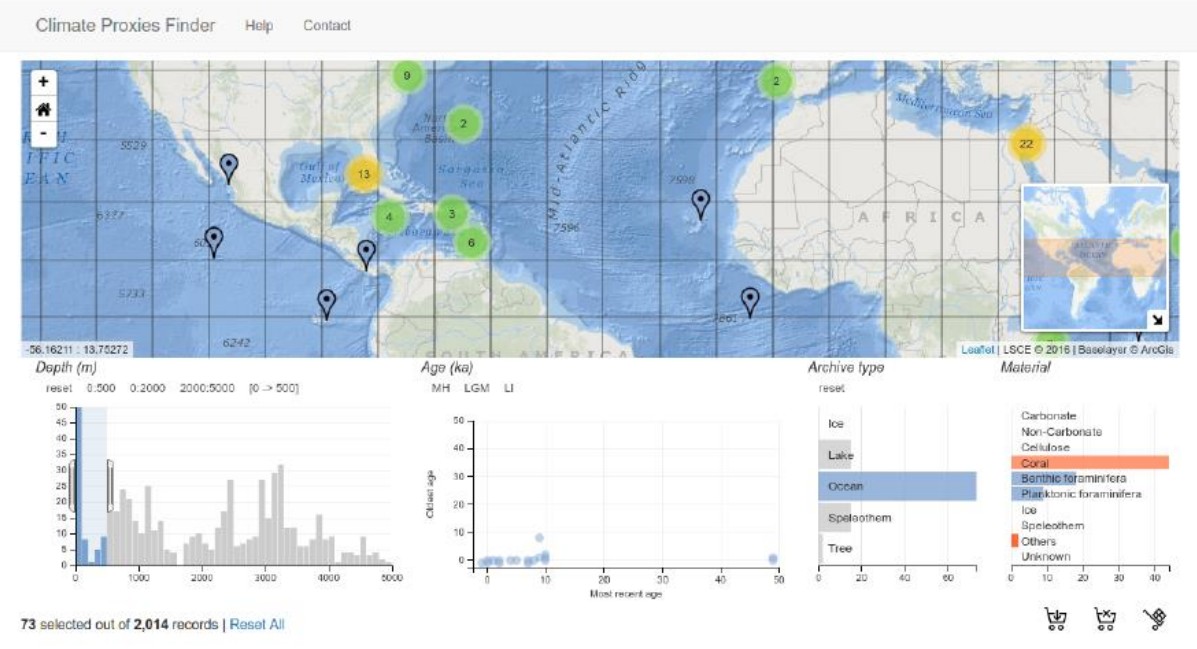



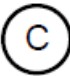

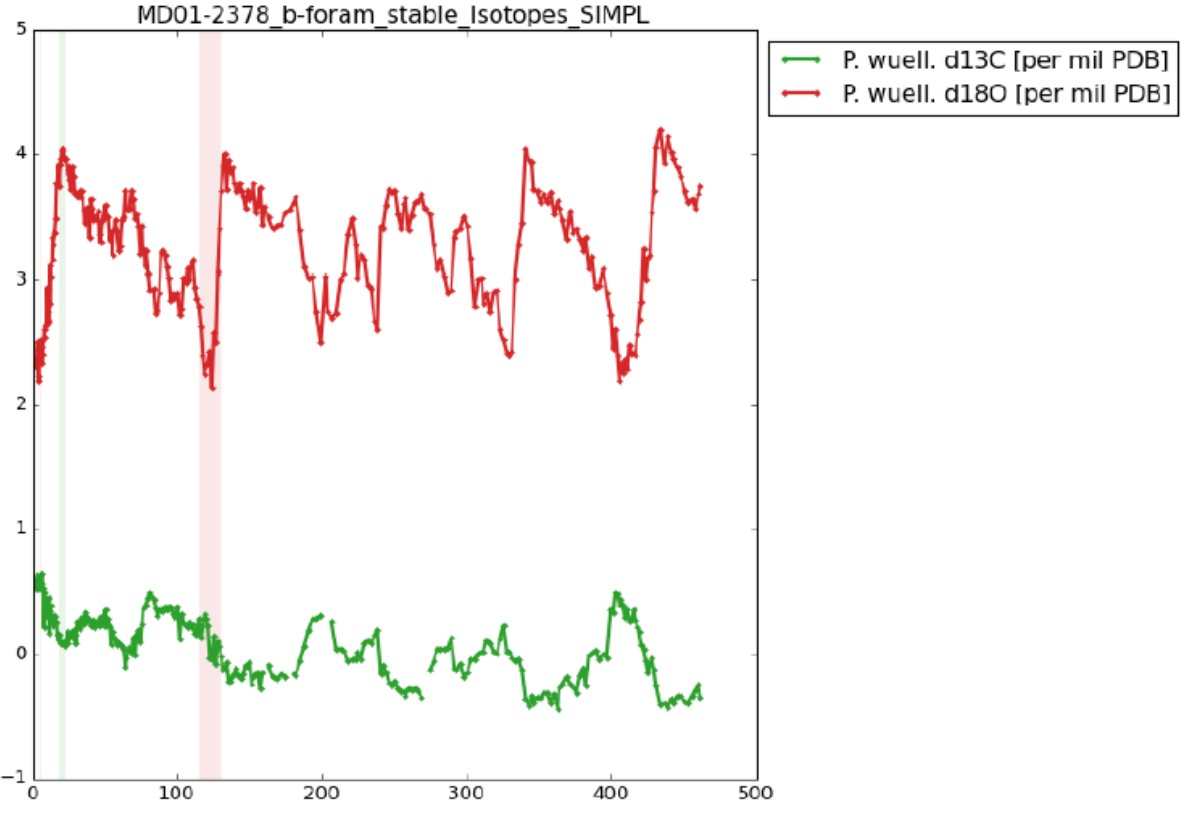


**Figure 2**

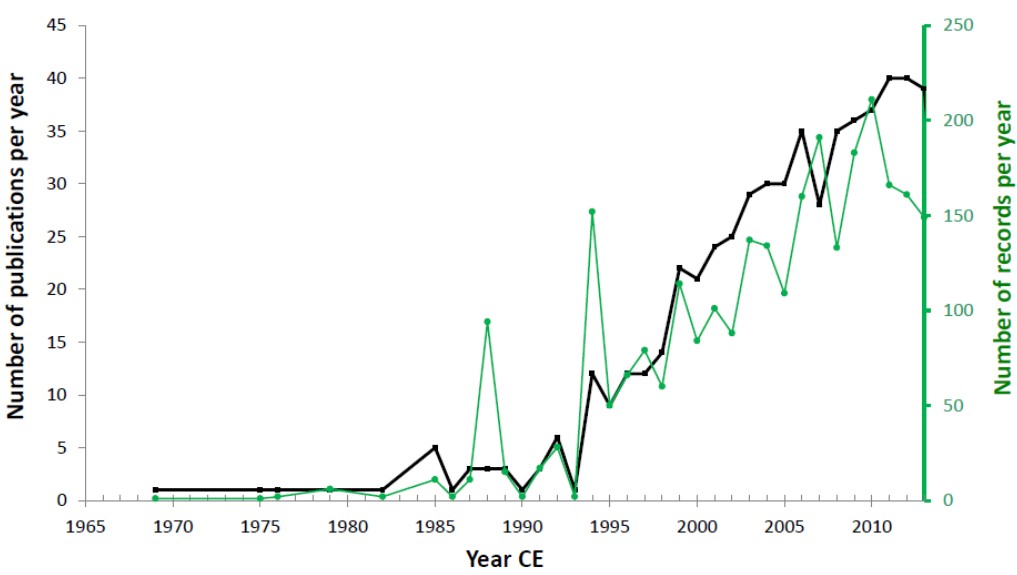



**Figure 3**

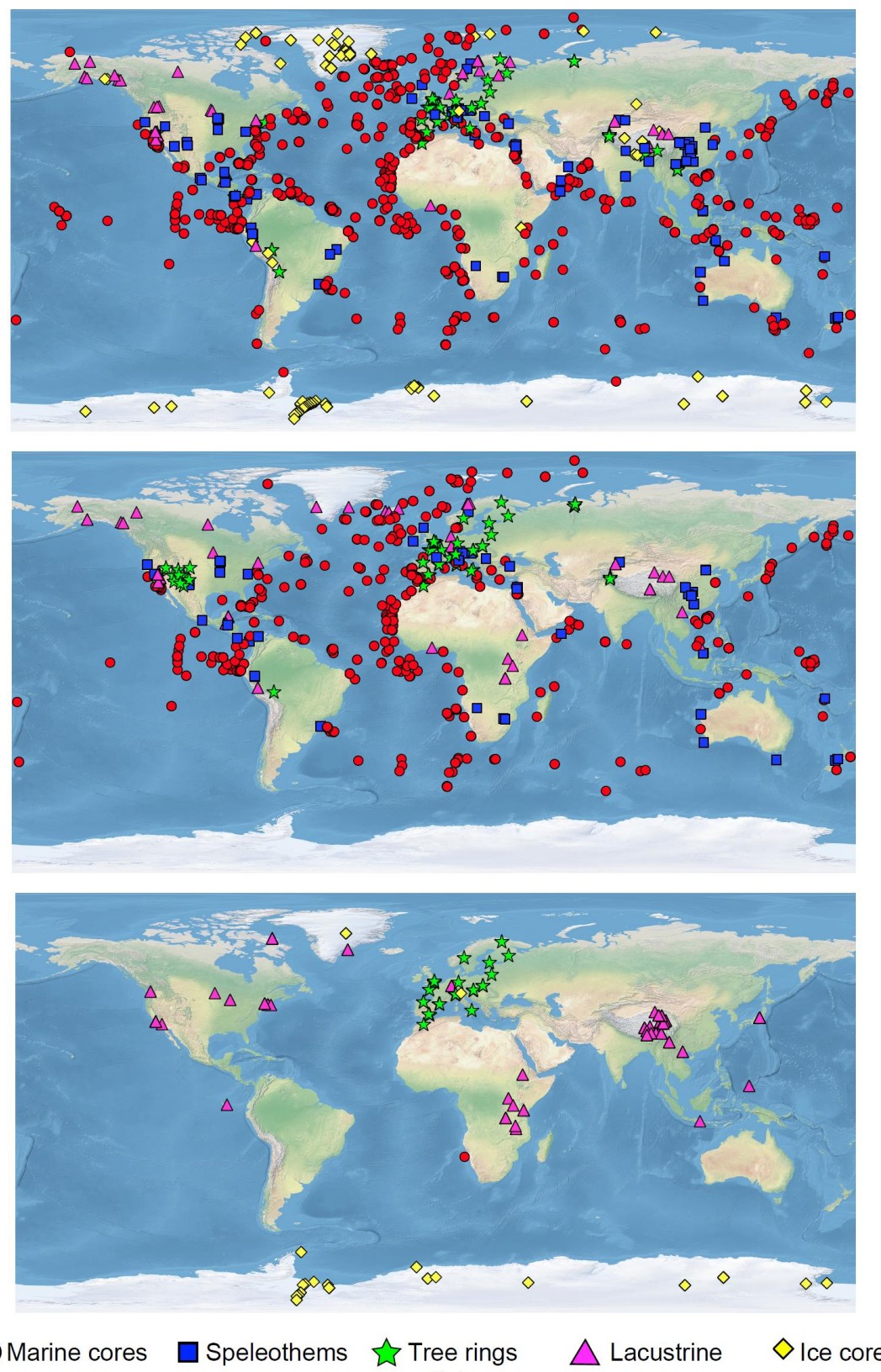

🔴 Marine cores  🟦 Speleothems  ⭐ Tree rings cellulose  🔺 Lacustrine cores  🔶 Ice cores


**Figure 4**

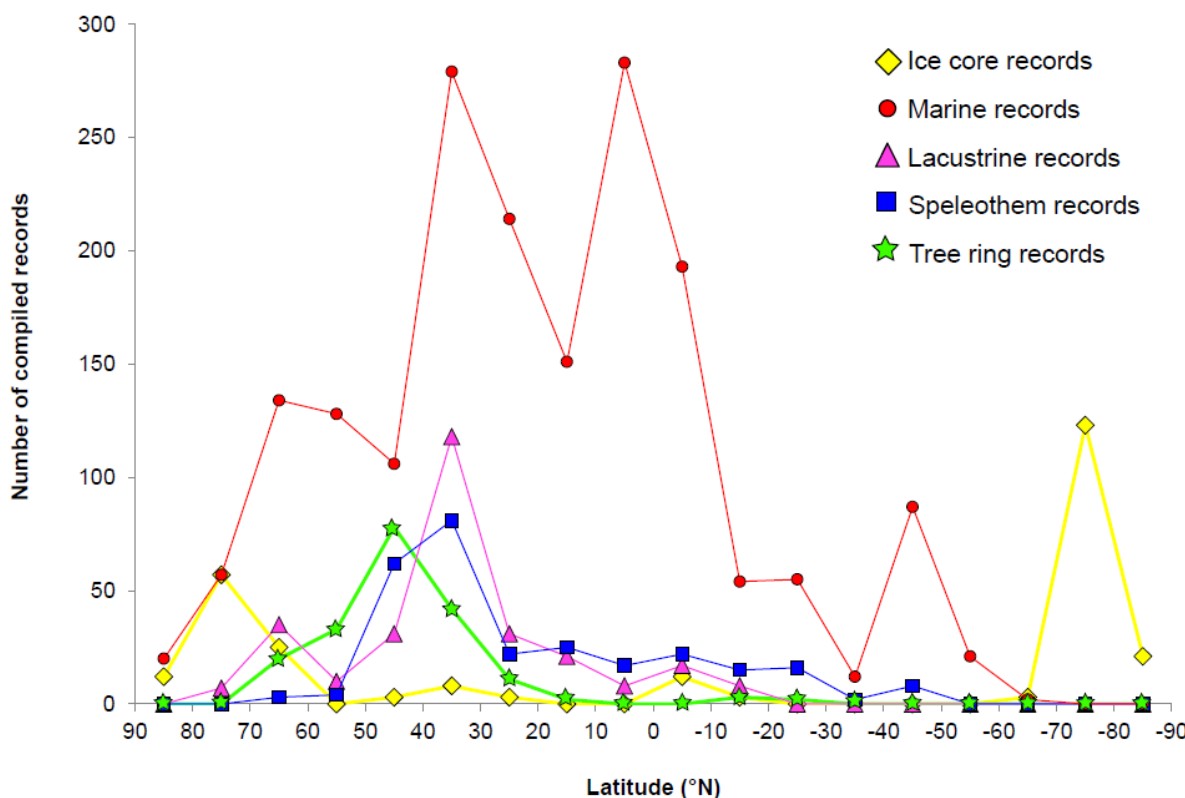



**Figure 5**

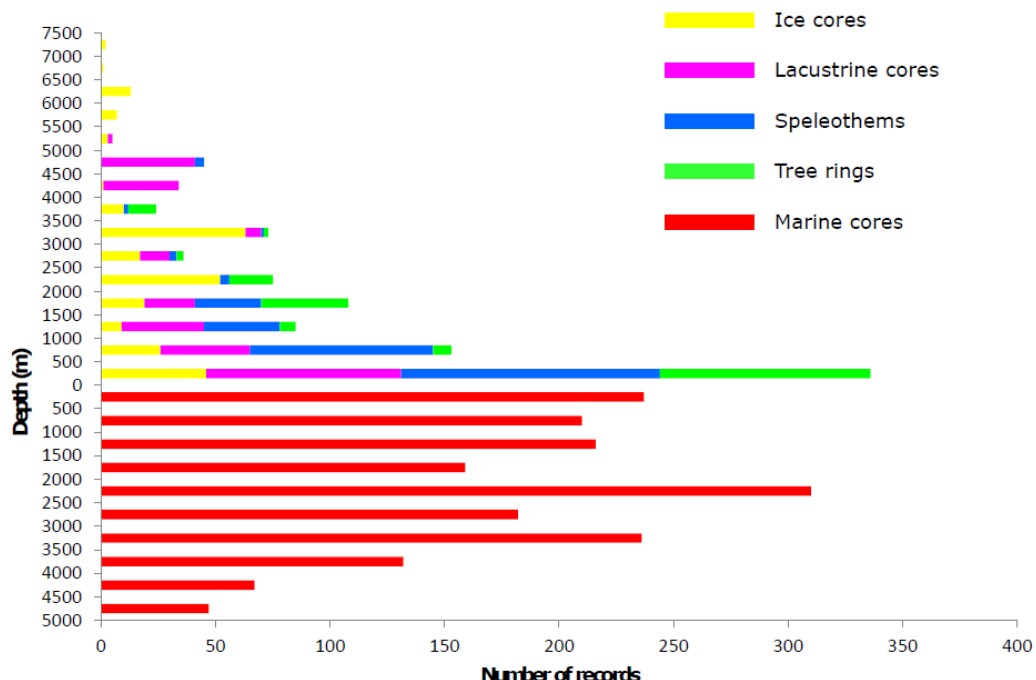


**Figure 6**

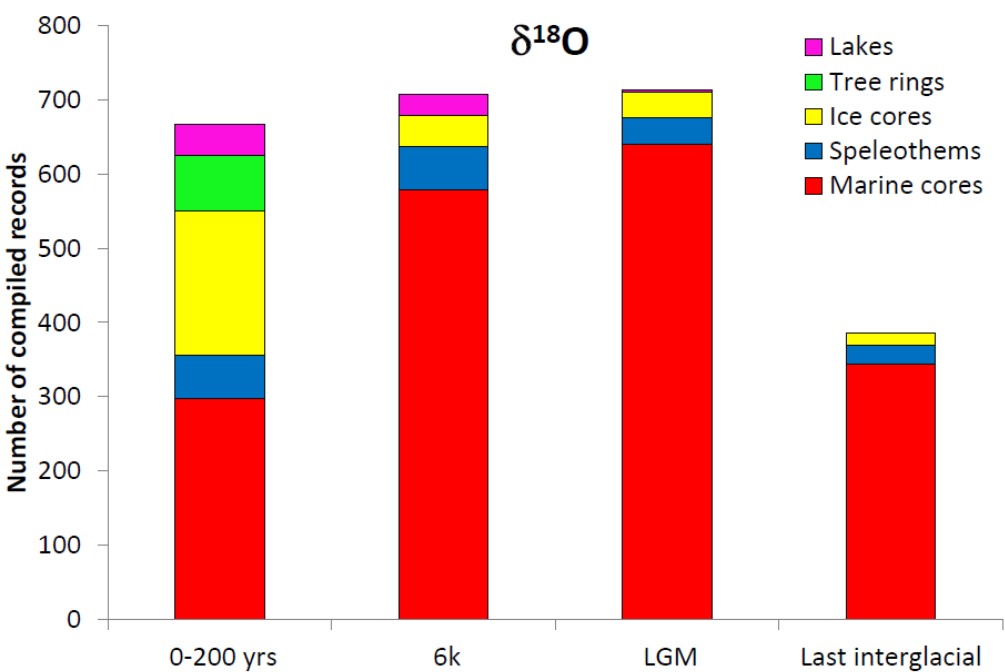


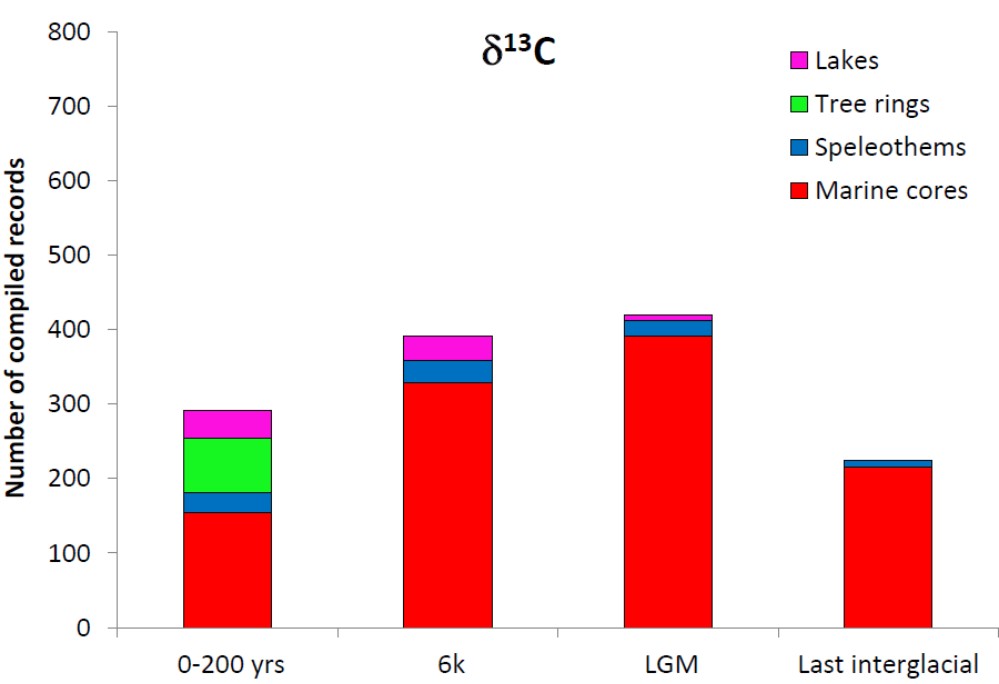


**Figure 7**

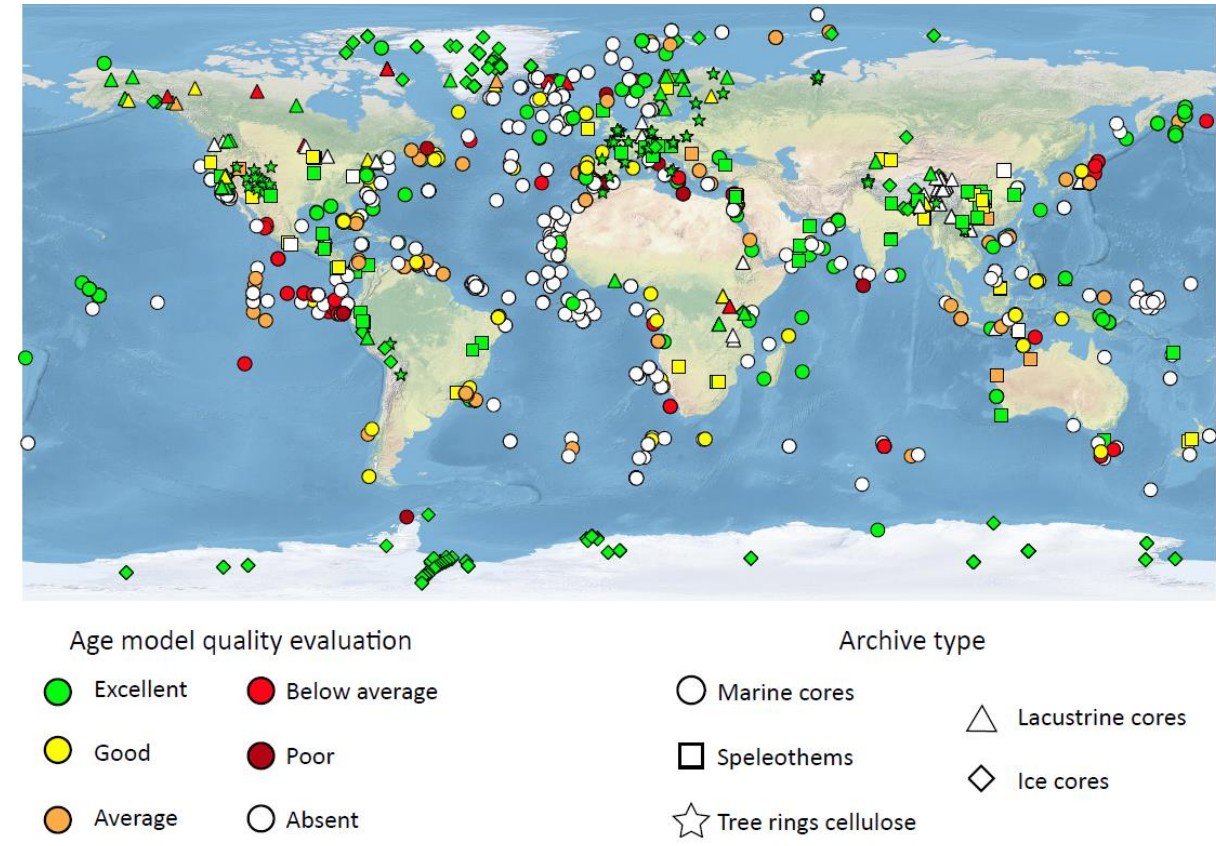


**Figure 8**

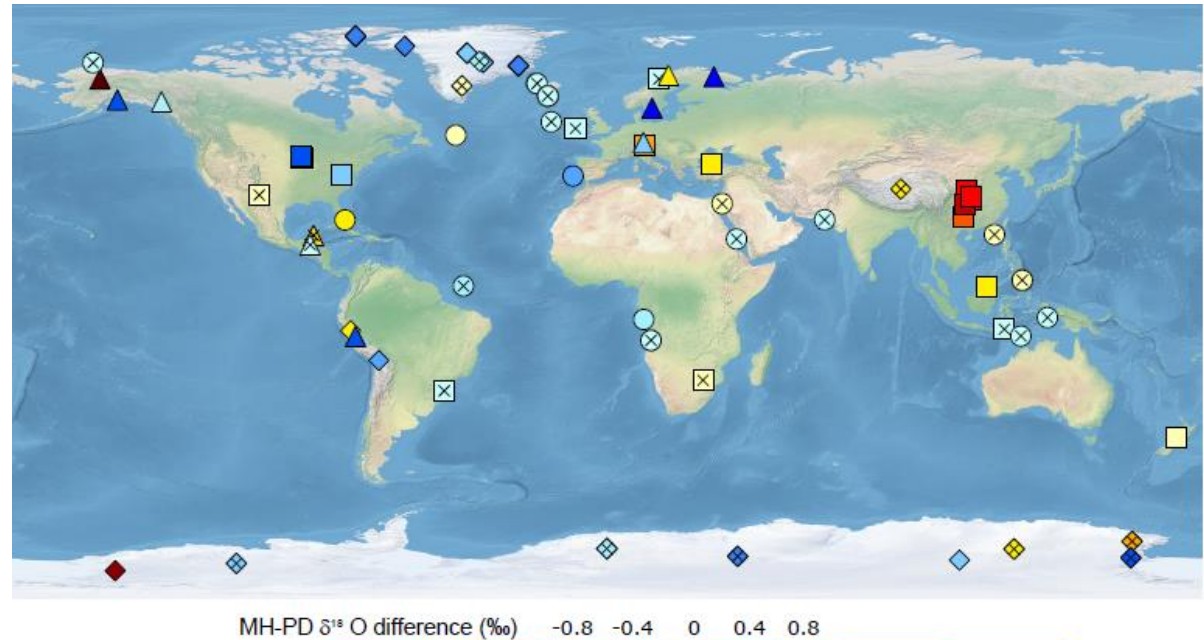

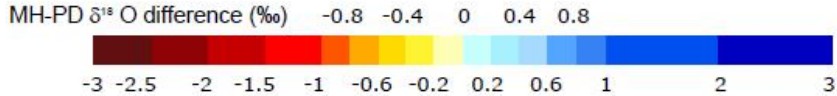

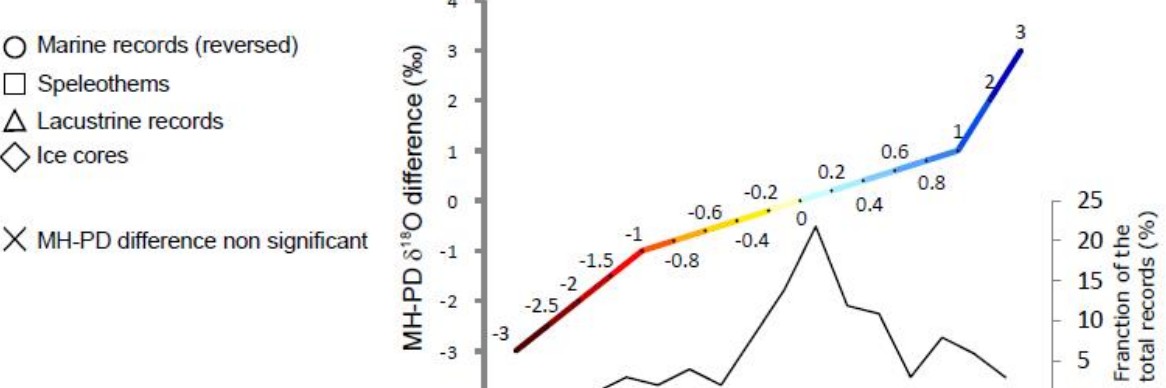



**Figure 9**

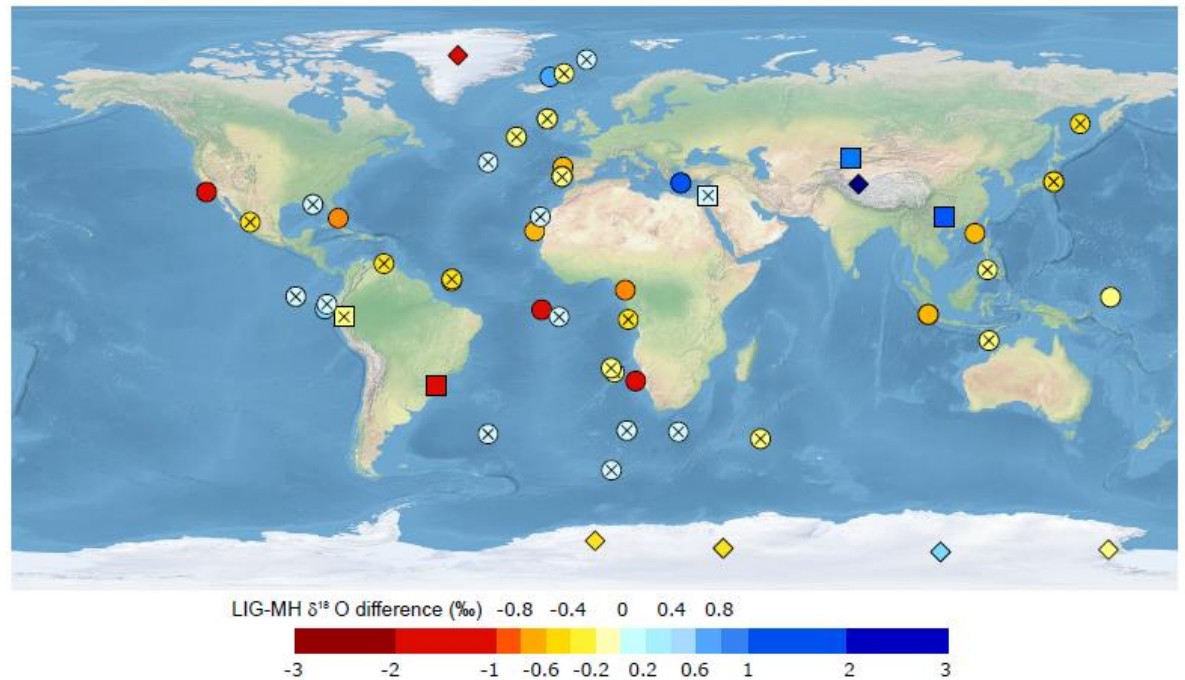

LIG-MH δ¹⁸ O difference (‰)   -0.8  -0.4   0   0.4  0.8

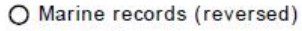

○ Marine records (reversed)
□ Speleothems
◇ Ice cores

✕ LIG-MH difference not significant

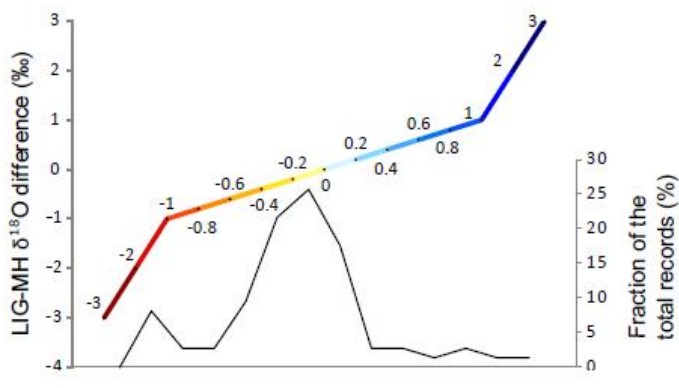



**Figure 10**

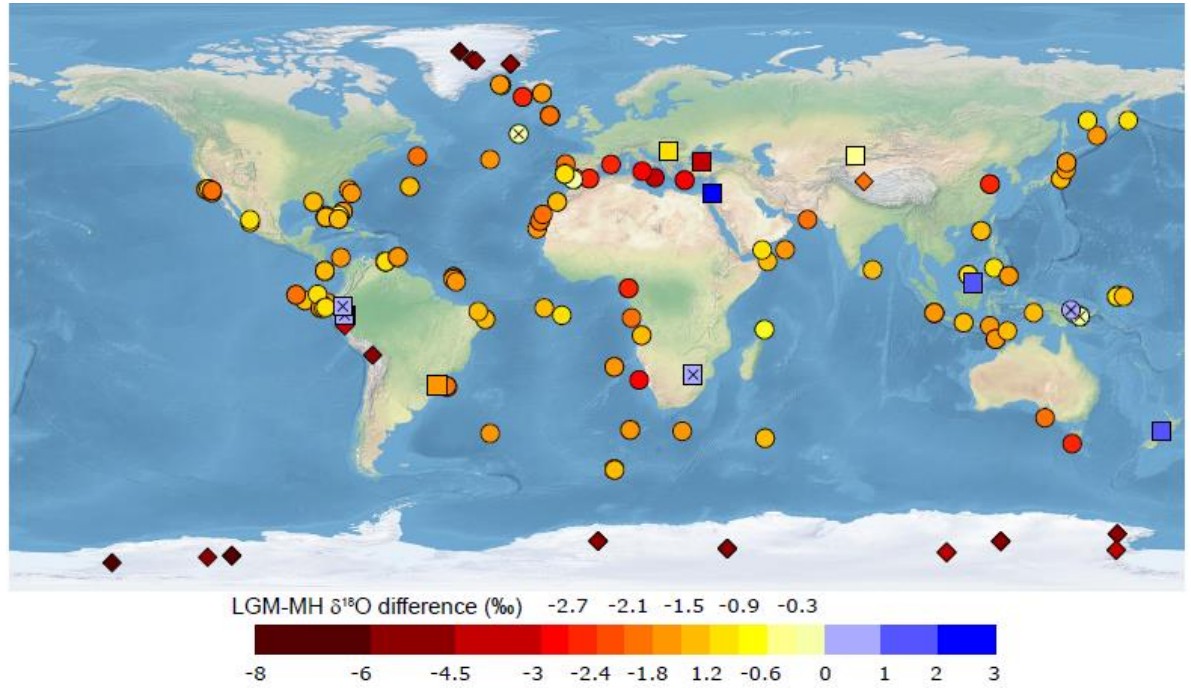

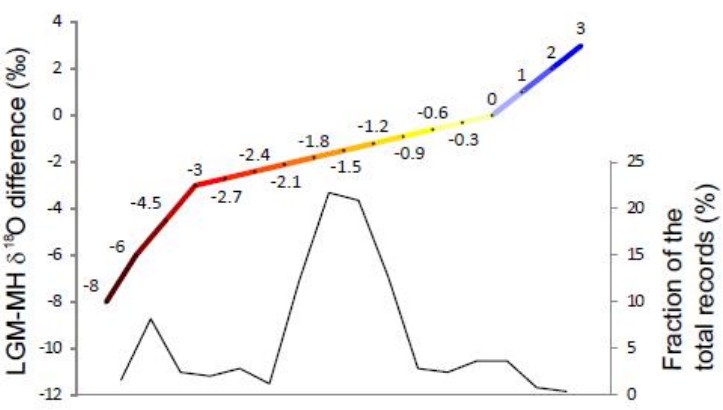




**Appendix**

**Statistical analysis – Estimation of the significance of the offset between PMIP time slices.**

The significance of the difference between two different PMIP time slices (A and B) was assessed by simply comparing the offset between the average isotopic value of these two periods ( $\overline{A}$ and $\overline{B}$), to the average value of the standard deviations of the isotopic record for each of the two periods ($\sigma A$ and $\sigma B$).

$$\overline{A} - \overline{B} \Leftrightarrow \frac{(\sigma A + \sigma B)}{2}$$

We consider that the isotopic offset is (not) significant if the absolute value of the offset is greater (smaller) than the average standard deviation along the two periods.

**Figure captions**

**Figure A1**: Diagrams showing the distribution of records (number of records) as a function of their mean time resolution (number of data points) for the different types of archives compiled in the database for the Present day (1800-2013 CE). Note the different vertical scales.

**Figure A2**: Same as Figure 1 but for the Mid-Holocene (5.5-6.5 ka). Note the different vertical scales.

**Figure A3**: Same as Figure 1 but for the LGM (19-23 ka). Note the different vertical scales.

**Figure A4**: Same as Figure 1 but for the last Interglacial (115-130 ka). Note the different vertical scales.

**Appendix Figures**

**Figure A1**

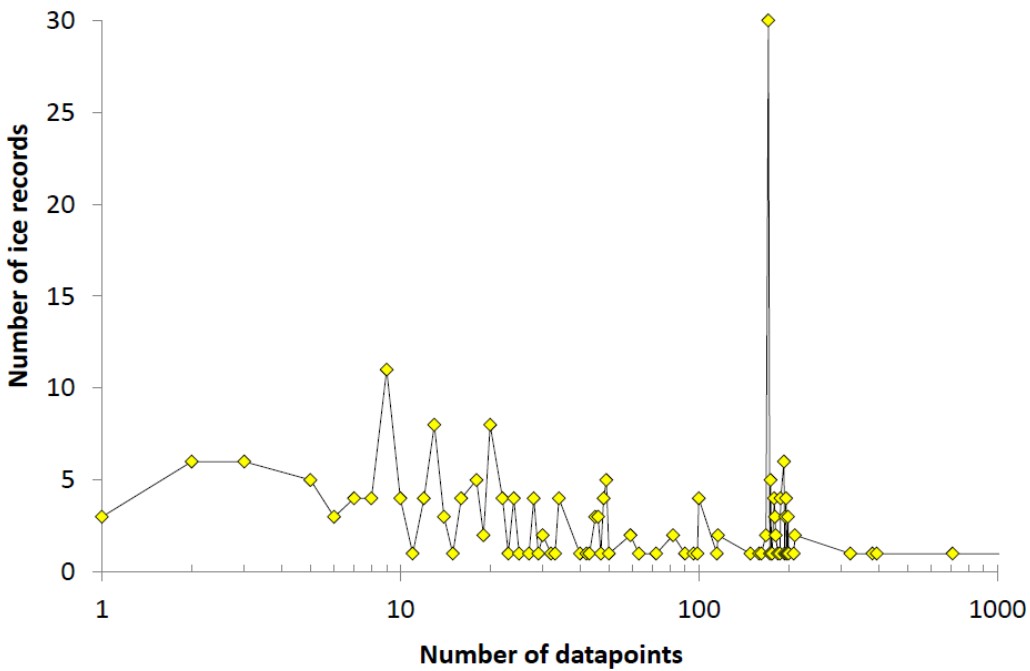


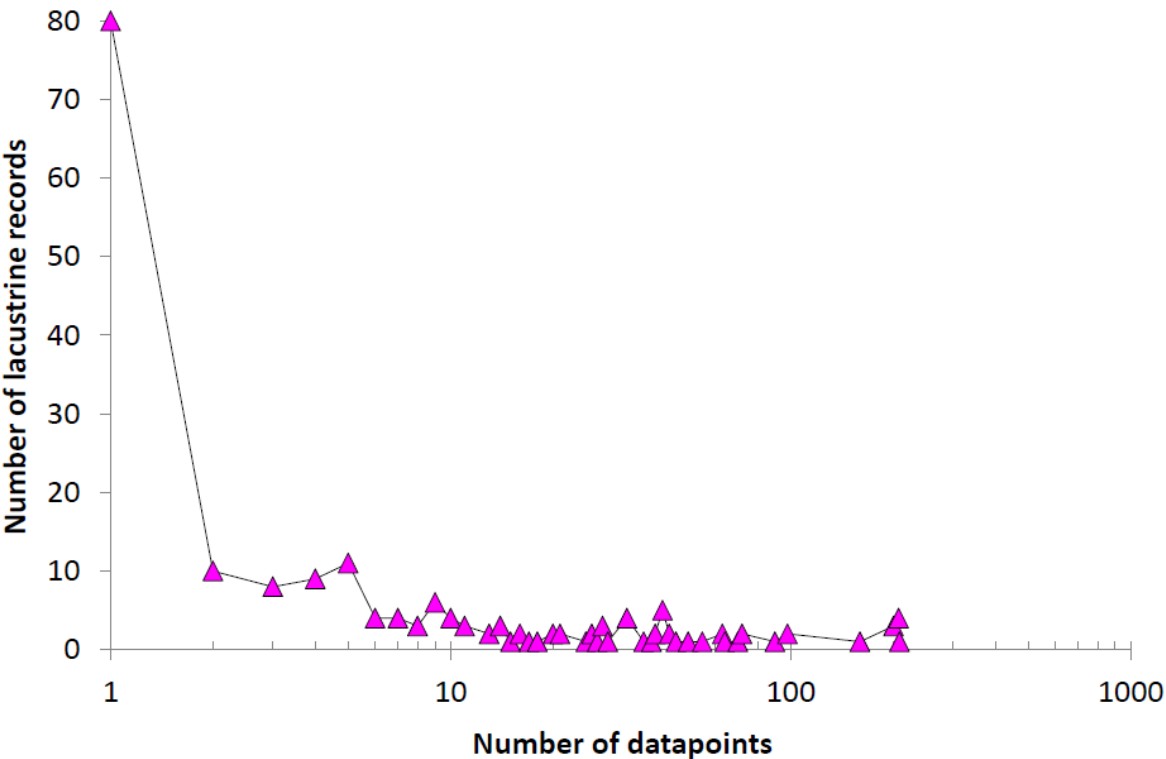


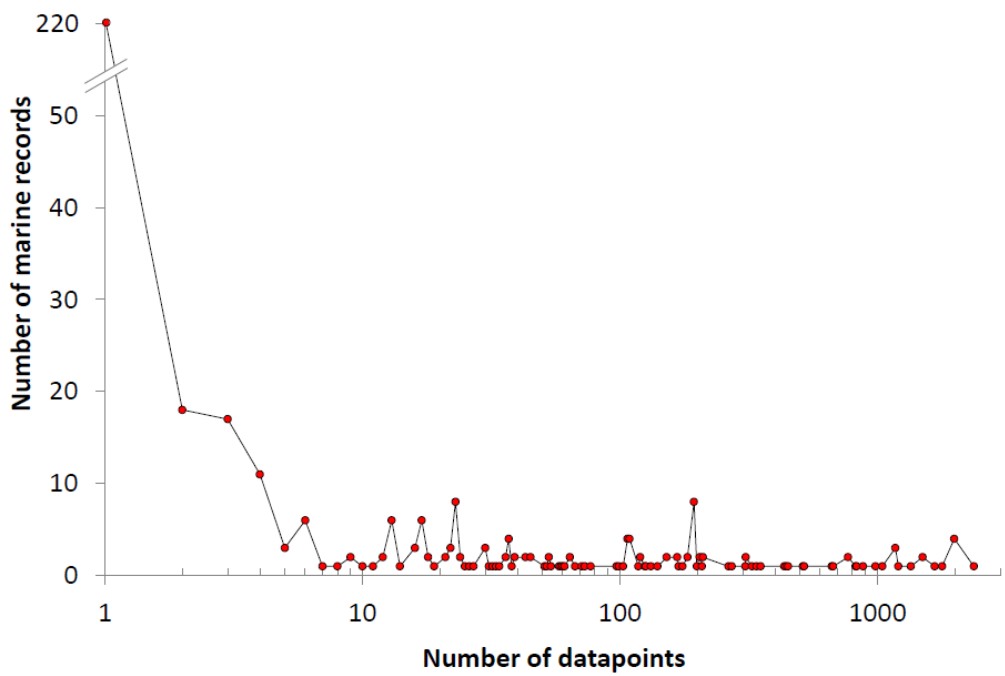


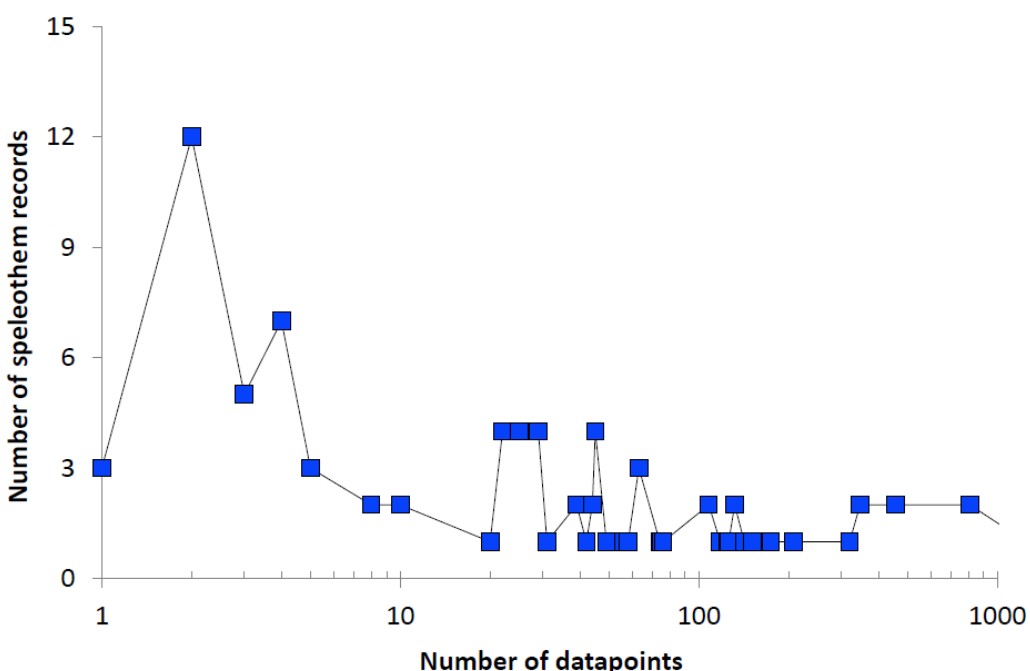


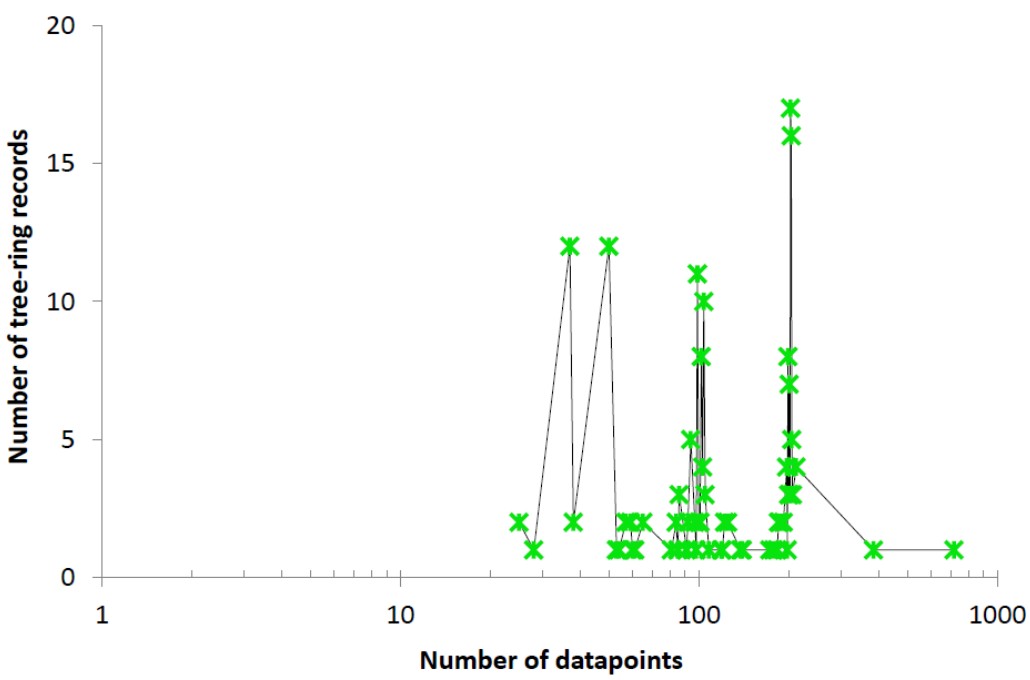



**Figure A2**

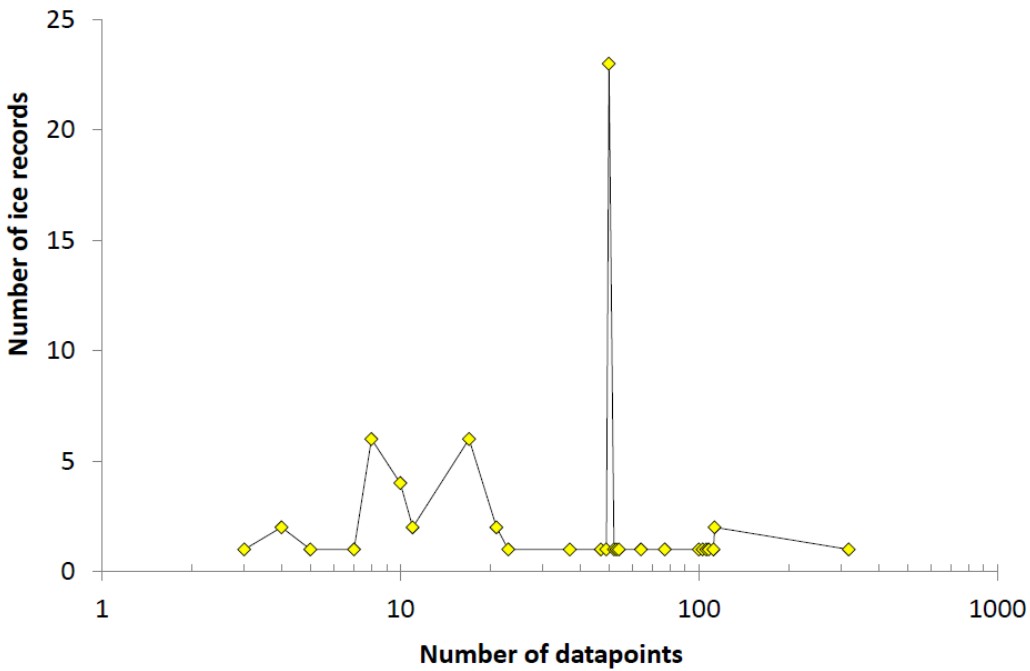


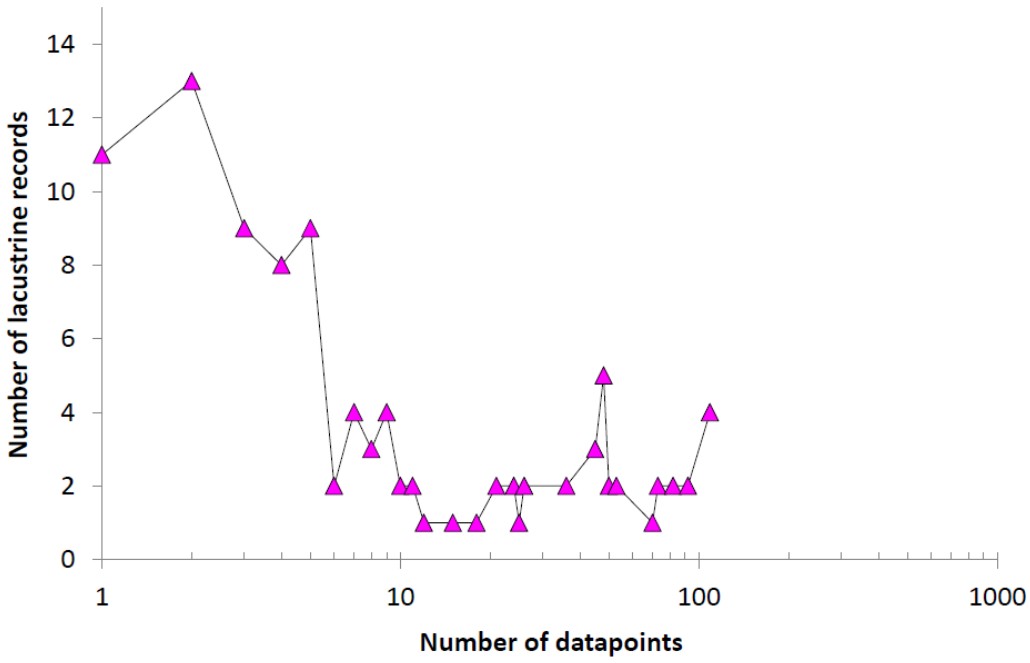


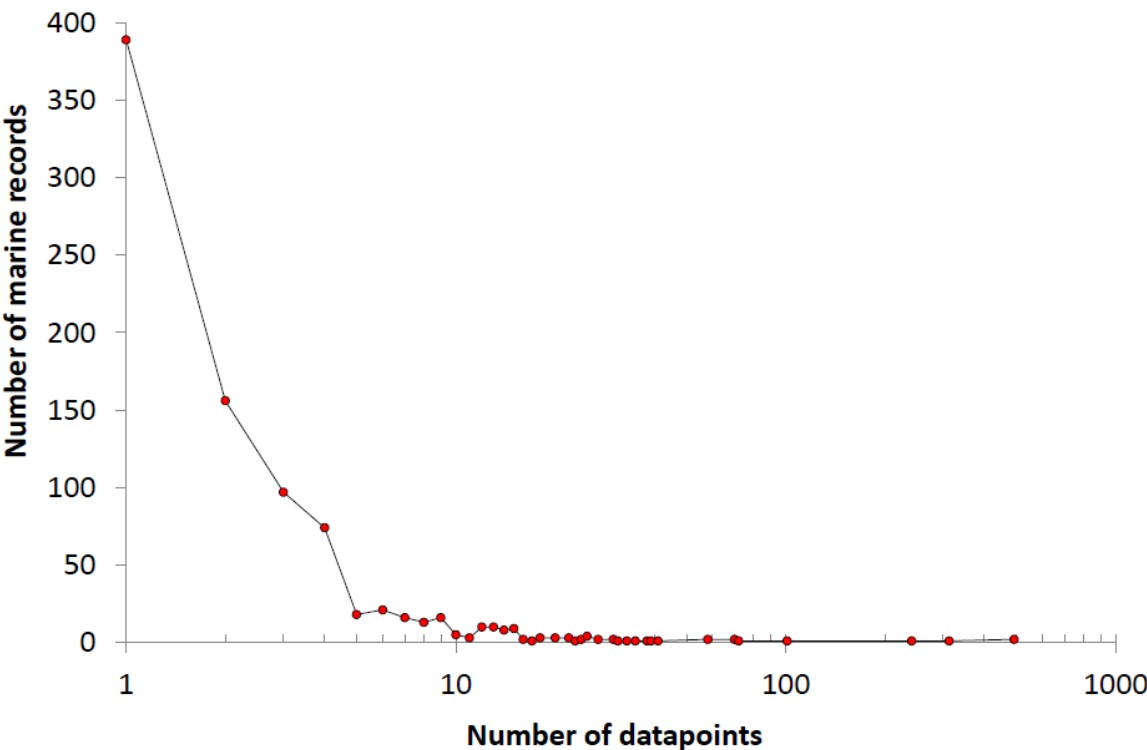


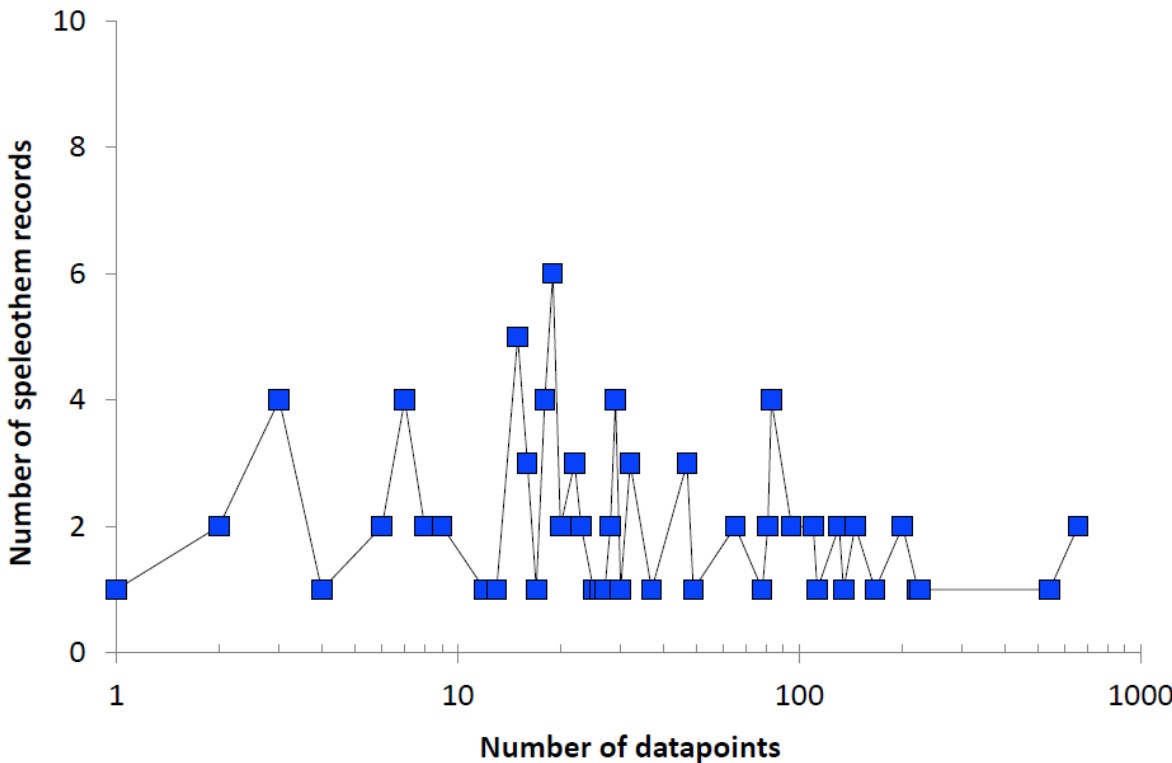


**Figure A3**

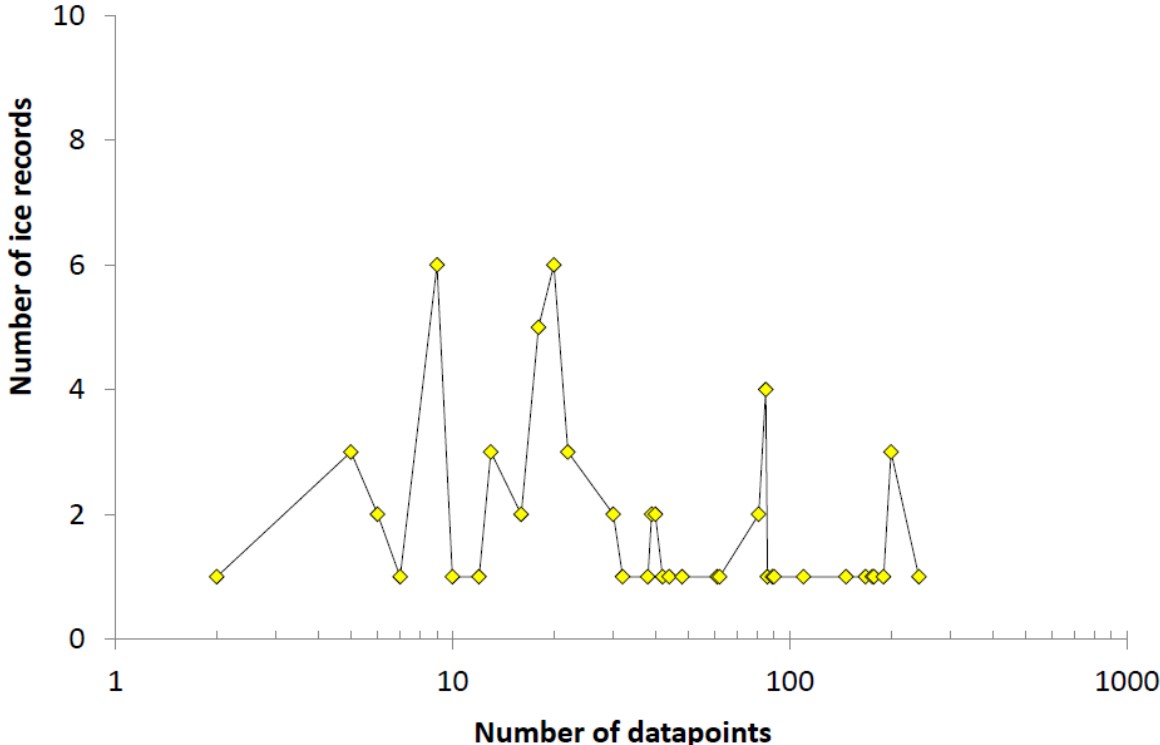


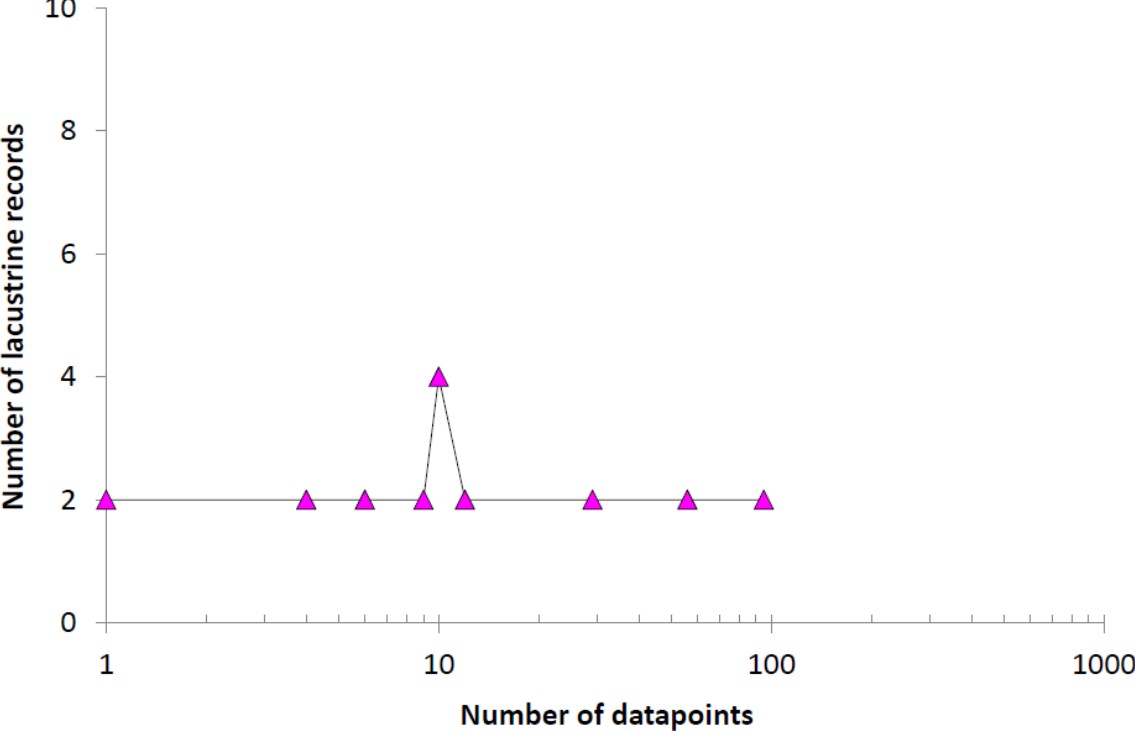


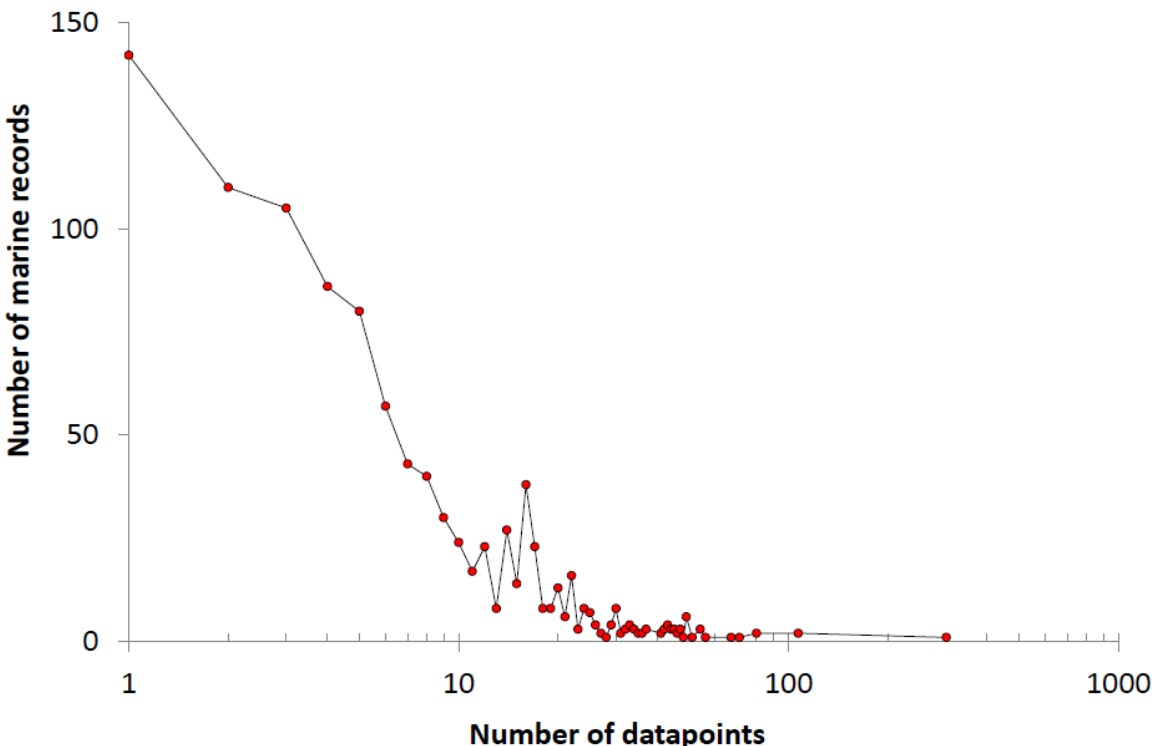


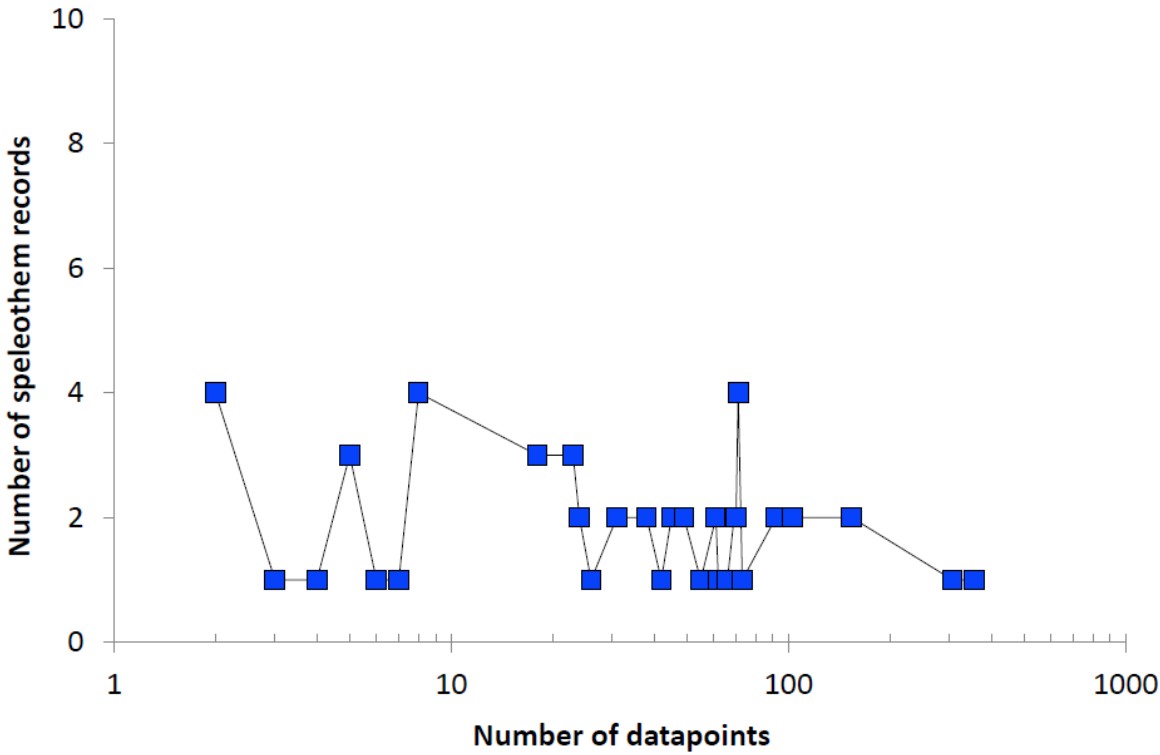


**Figure A4**

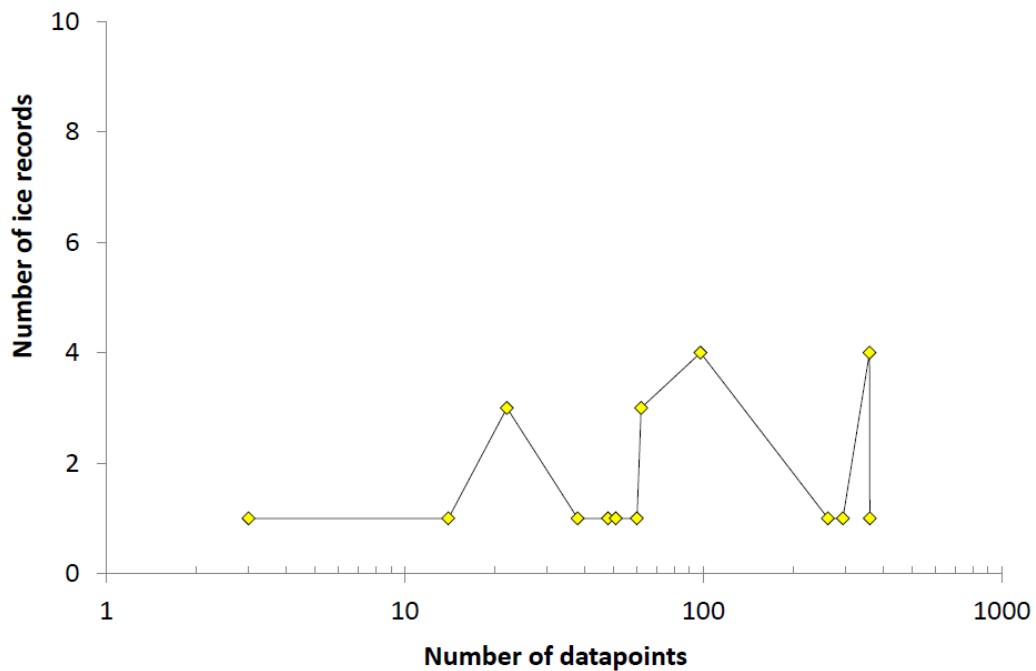


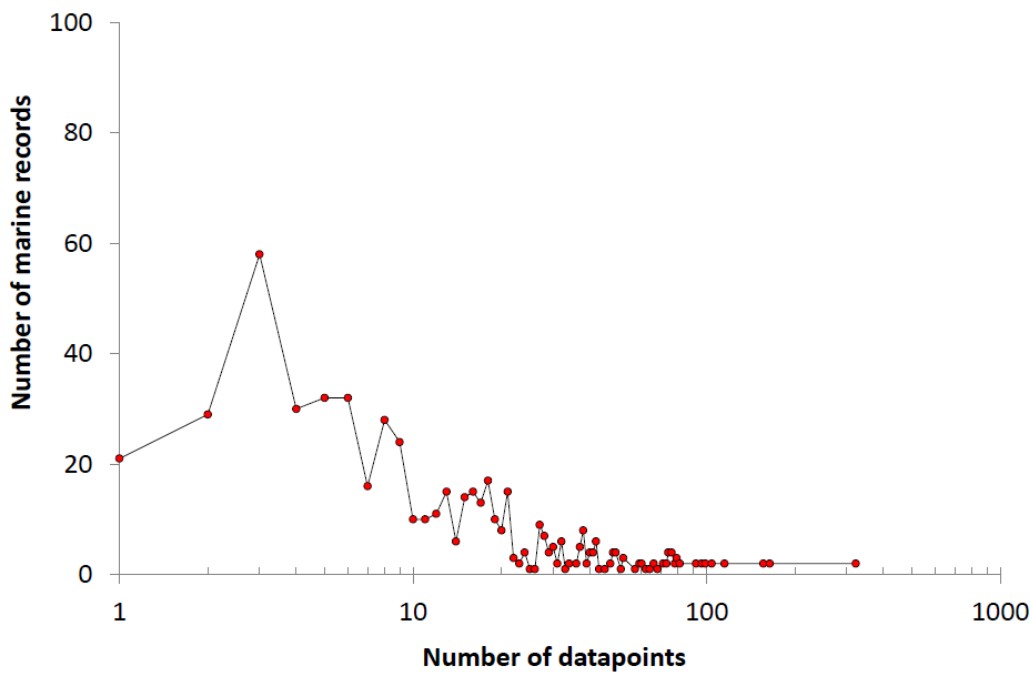


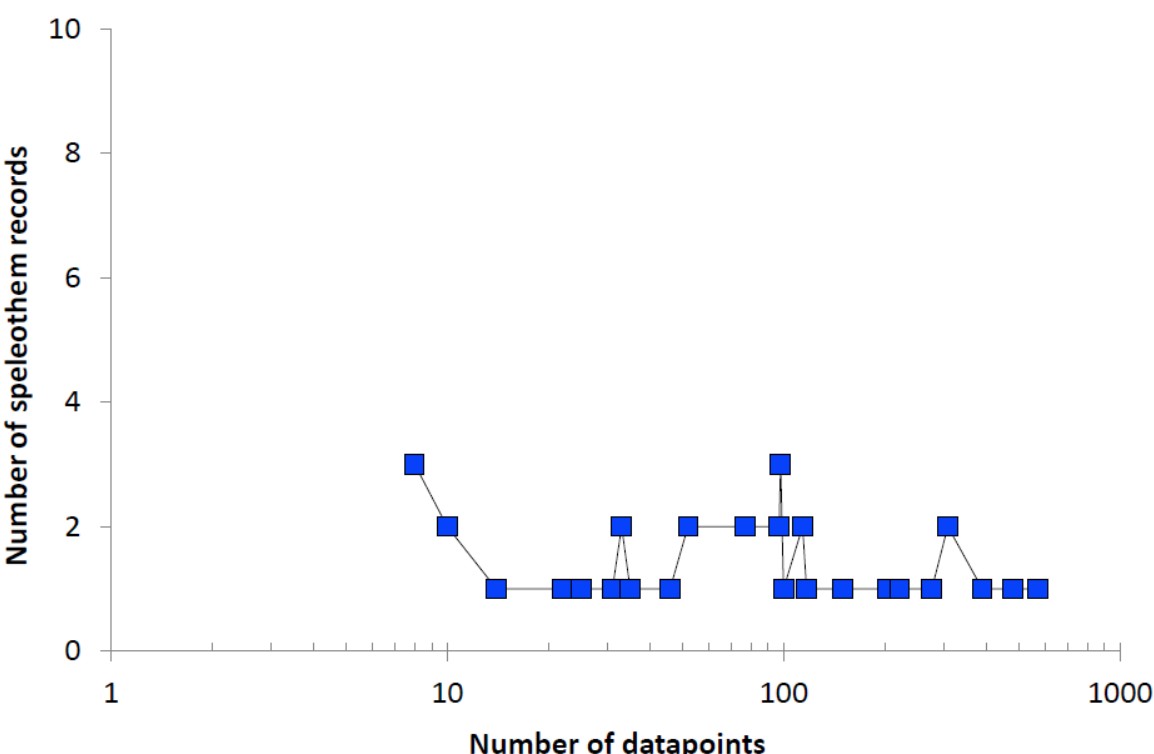
