# Peer review of "Water and carbon stable isotope records from natural archives : a new"

_Climate of the Past, 2015_

## Referee Comment (RC1) · Anonymous Referee #1 · 7 Mar 2016

The manuscript of Bolliet et al. describes a new compilation of isotope proxy records (oxygen, hydrogen, carbon), which are key for many paleoclimate studies. According to the authors, they have compiled several hundreds of records from various archives (marine and lacustrine sediments, ice cores, tree rings, speleothems, corals). So far, these records have been either published in the literature or scattered among different other databases, e.g. NOAA libraries. This new compilation focuses on four different time periods of the past, only: (i) the last 200years, (ii) the mid-Holocene, (iii) the LGM, (iv) the last Interglacial. With this compilation, the authors aim to homogenize the format of the available data sets and enable an easier usage and wider distribution of the isotope data via the online portal http://climateproxiesfinder.ipsl.fr. This online portal

shall enable scientists to easily search and find specific subsets of different criteria within the compilation and also download the data in a homogenized format.

I rate the efforts of the authors to create such a new data compilation as worth publishing and highly valuable for the scientific community. Nowadays, more and more research time is consumed for screening the growing amount of literature and data sources, finding relevant data sets and getting them into a useful common format for own studies. Compilations like the one described by the authors might help to perform these tasks in a more efficient manner. Thus, I highly support the compilation and publication of this database (even if it is rather specific and focuses on a few selected proxies and time slices, only) and compliment the authors on their efforts.

The manuscript is well outlaid and written in a clear and concise manner. It can almost be published it its present form. However, I am still hesitating to recommend a publication in Climate of the Past for two reasons:

(i) The authors describe a database, which is not available to the public, yet. As of today (March 7, 2016), the website http://climateproxiesfinder.ipsl.fr is linked to the compilation version 1.45, updated on 2016/02/26. Graphical data presentation and access via this website look very different from the version described by the authors, and an option for downloading data in a common format does not exist, yet. Thus, I have currently no possibility of checking key statements of the manuscript about data availability as well as data selection and download options.

(ii) The article is merely a description of the new data compilation and built website interface. In my opinion, it does not contain enough new scientific insights to merit a publication in CP. Other journals (e.g. Earth System Science Data ESSD) might be much more appropriate for such kind of article. However, I realize that CP has published similar compilation efforts in the past.

Because of these demurs, I leave the final decision about the appropriateness of publication with the editor.

---

## Referee Comment (RC2) · Anonymous Referee #2 · 28 Mar 2016

**Recommendation**: *Accept after major revisions*

[Figure]

Summary The article presents an online database of paleoclimate records suitable for PMIP-style time-slice investigations of past climate change. The paper presents the data synthesis, the online navigation tool, and some elementary analyses enabled by the dataset and platform. Overall the effort is laudable, but important philosophical and technical issues prevent publication in the current form. In particular, a lack of development on the data standards front severely limits the scope and ambition of the current work.

**1 Scientific Comments**

**Unclear goal:** the goal of the paper is somewhat unclear. Is it to present the interactive tool, which is the main subject of the title? If so, why does the paper spend most of its time on a description of the database? To me, the efforts are data standardizations presented here are quite preliminary, and the most unique and interesting part of the paper is the online portal, which should be further described.

**Generalizability:** Accordingly, are the software tools developed here (by far the most interesting aspect of the paper) open-source? Can other institutions (e.g. NOAA, PANGEA) benefit from this investment of programmer time? Right now there is no mention of a code repository (e.g. GitHub). It would seem contradictory for an open-access journal like Climate of the Past to promote work that is not open-source.

**Data standardization:** The authors rightly "highlight the needs for a standardized protocol of data storage". The thing is, such a protocol already exists (Linked Paleo Data, or LiPD), and the paper describing it is in press in the very same journal (discussion paper: *McKay and Emile-Geay*, 2015). In fact, the LiPD data

container is more comprehensive than what the authors describe, is machine-readable, web-searchable, and designed to carry all the metadata together with the data, so it would solve many of the problems raised here. The authors should mention such efforts in a revised version of the manuscript.

**Data synthesis** : The authors acknowledge in the discussion that MARGO-style data syntheses are of limited interest because they become obsolete almost as soon as they are published. The main value of their product, then, is to enable a system whereby the database can grow organically over time. The authors "highly encourage authors to upload their new data in our database using the user-friendly interface on the online platform." Is the platform meant to replace NCDC Paleo or PANGAEA? If not, how will it ensure a steady flow of records from those repositories to the platform? The LiPD-based LinkedEarth project (http://linked.earth) shares similar features, such as the ability to not only upload new datasets, but edit existing ones to correct errors or expand metadata. Importantly, LinkedEarth involves a partnership with NCEI to harmonize the two databases. A lack of harmonization would result in duplicate (and sometimes, conflicting) information in cyberspace, which may do more harm than good. I invite the authors to take a look at the LinkedEarth framework and see how it could mesh with their efforts.

**Age modeling** : It is somewhat disappointing that the authors focused only on "time slice" investigations, but I gather from the discussion that they are thinking more broadly, and I surmise that the science driver behind this data organization effort must be the PMIP project. Indeed, PMIP and other paleo projects should move towards evaluations of transient change, so it is critical that all age constraints be archived in a standard format. This was, in fact, the main motivation behind the creation of the LiPD standard. The authors are correct that there is no accepted standard for how to store age information for any archive. Even in the radiocarbon community, practices vary by country, lab or institution. A very useful contribution of the paper would be to list all the data columns necessary to reproduce an age

model (e.g. OxCal, BACON, BChron), since this should be the science driver: ensure that age modeling tools will be able to make use of the records archived in the database, by focusing on what information they need. The authors mention reservoir ages, which are essential. Just as importantly, the *uncertainty* about reservoir ages is absolutely fundamental, and neglecting it can result is grossly overconfident assessment of ages. It would be an important contribution if the paper made this clear, and proposed a format for how to report such ages and their uncertainties. The existence of inter-lab in radiocarbon dates of identical samples also makes it necessary to include lab (and a sample ID) as essential metadata to keep track of.

The authors also acknowledge that layer-counted records also harbor age uncertainties, though they seem to neglect them. That approximation is certainly justified for the time slices they consider here, but since they are building a case for how to more properly archive paleoclimate records, I think they should mention the work of *Comboul et al.* (2014) (also in this venerable journal) and its call to report the probabilities of under-counting and over-counting layers, since these are necessary ingredients to modeling the uncertainties in such records.

**Chronology ratings** : The authors devised a 5-point scale to grade the quality of chronologies. This seems like a useful semi-quantitative criterion that users will want to select records based on the requirements of their analyses. However, I was disappointed that the criteria are so qualitative. The only quantitative criterion involves the density of tie points, not the quality of the measurements. Can the scale be based on objective measures, like the width of the posterior distribution of ages (as measured, for instance, by a 95% highest-density region or the inter-quartile range)? It would seem less controversial to do this, otherwise I can envision endless arguments about the ratings. For instance, why "consider that the LGM (19–23 ka) is better constrained with two AMS dates at 19 and 23 ka than with four dates within the 20–22 ka interval"? This seems arbitrary to me.

**Statistical Analysis** : in section 6 the authors analyze differences between isotopic records for various time slices. However, there is no mention of the test that was used to determine significance. If it was a t-test (as seems likely), did it consider the reduction in the number of degrees of freedom due to the low-resolution? (i.e. having 10 points over the past 200 years may not been that there are 10 degrees of freedom going into the estimate of the mean). The authors need to explain in detail which methodology was used for this analysis. The results are quite interesting, but they are meaningless without more thorough methodological information.

**2 Editorial Comments**

- Eventual: this adjective, used in 3 instances, seems like a Gallicism for "potential" or "possible". In English, eventual means final, which is not the intended meaning (I think).

- "Model outputs" is used a few times, whereas "model output" (non-denumerable) would be grammatically sound.

- P4: L8: "Past climate was the result...". Awkward entry. How about "Past climate variations resulted from ... "

- P5, in the enumeration of isotope-enabled GCMs, please add SPEEDY-IER, (*Dee et al.*, 2015b)

- P5, L13 "These new functionalities of climate models open the possibility to directly comparing the proxies measured in natural archives with model outputs": this is not correct. Though isotope-enabled GCMs bring us a step closer to closing the proxy-model gap, proxy systems impose many more transformations of

the environmental signal, and they must be taken into account , e.g. via proxy system models (*Evans et al.*, 2013; *Dee et al.*, 2015a).

- P6, L10: depositories –> repositories

- P7, L3: "integratation" –> integration

- P7, "The last 200 years (1800 to 2013 CE, Common Era) ": technically, this intervals comprises 214 years.

- P10, L26: "simple text tabulated standard format". actually this is anything but simple, as tabs are not encoded the same way by every machine. Also, how are missing values encoded?

- P15,L12: "2s error" : do you mean $2\times$ the standard error of the mean?

- P15, L20, how are the layer-counting errors reported? They are measurable, and sometimes larger than radiometric dates (*Shen et al.*, 2013)

- P16, L11: affected by few –> affected by a few

- P18, L19: "a few $\delta^{17}O$ records": How many?

- P21, L4: less than –> fewer than

- P23,24: this is far too descriptive, and very redundant with section 4. I recommend condensing or cutting if possible.

- P30, L14: "instauration of standard time units" . Firstly, instauration is not in common use in English. Consider "establishment". Secondly, I am not sure this proposition is wise, because paleogeoscientists develop records for very different purposes and the time standard depends on the time frame (e.g. Common Era

vs Jurassic). Instead, what is needed is a more flexible data standard (like LiPD), that can accommodate many conventions.

FIGURES

- Fig 5 misses a legend about the meaning of the colors. (I know it is consistent with other figures, but it would be helpful for each figure to be self-contained).

- Fig 7, same thing: repeat the meaning of symbols for each archive type.

- Fig 8 : need a colorbar to explain colors.

- A1–4: consider binning on the x-axis for clarity. Right now they look a bit too spiky to be useful.

In summary, this is a valuable contribution, and I recommend publication after these points have been addressed.

**References**

Comboul, M., J. Emile-Geay, M. N. Evans, N. Mirnateghi, K. M. Cobb, and D. M. Thompson (2014), A probabilistic model of chronological errors in layer-counted climate proxies: applications to annually banded coral archives, *Climate of the Past*, *10*(2), 825–841, doi: 10.5194/cp-10-825-2014.

Dee, S., J. Emile-Geay, M. N. Evans, A. Allam, E. J. Steig, and D. M. Thompson (2015a), PRYSM: An open-source framework for PRoxY System Modeling, with applications to oxygen-isotope systems, *Journal of Advances in Modeling Earth Systems*, *7*(3), 1220–1247, doi:10.1002/2015MS000447.

Dee, S., D. Noone, N. Buenning, J. Emile-Geay, and Y. Zhou (2015b), SPEEDY-IER: A fast atmospheric GCM with water isotope physics, *Journal of Geophysical Research: Atmospheres*, *120*(1), 2014JD022,194, doi:10.1002/2014JD022194.
Evans, M. N., S. E. Tolwinski-Ward, D. M. Thompson, and K. J. Anchukaitis (2013), Applications of proxy system modeling in high resolution paleoclimatology, *Quaternary Science Reviews*, *76*(0), 16–28, doi:10.1016/j.quascirev.2013.05.024.

McKay, N. P., and J. Emile-Geay (2015), Technical note: The linked paleo data framework: a common tongue for paleoclimatology, *Climate of the Past Discussions*, *11*(5), 4309–4327, doi:10.5194/cpd-11-4309-2015.

Shen, C.-C., K. Lin, W. Duan, X. Jiang, J. W. Partin, R. L. Edwards, H. Cheng, and M. Tan (2013), Testing the annual nature of speleothem banding, *Scientific Reports*, *3*, 2633 EP –, doi:10.1038/srep02633.

---

## Author Comment (AC1) · 10 May 2016

The first aspect highlighted by Reviewer #1 is the fact that the online portal was not fully functional at the time of the review process. We have been working on it during the last weeks and users can now perform data browsing, visualization and download. We are however still working on it to build the upload feature. These are the four functions included in the original project, but we are highly open to suggestions and collaborations concerning the implementation of additional features to our online portal.

Reviewer #1 also mentioned the fact that it may be more appropriate for this article to be submitted to a "data-orientated" journal. We however estimate that our article is addressed to various data users, especially when we highlight the urgent need to find

an agreement for the format of metadata and age modelling data. This issue needs to be discussed by a large panel of palaeoclimatologists, including data producers, modelers, gatherers, and we consequently believe that it would be less appropriate to submit it to a somehow specialized journals that may be addressed to a more restricted and specific audience.

———————————————

---

## Author Comment (AC2) · 10 May 2016

Scientific comments

Unclear goal

Thank you for stressing this issue, and we have done our best to clarify our goals in the revised version. This paper had indeed two main original goals. The first one was to provide a simplified and homogenous format for the isotopes data published without any common format or protocol during the last decades, and to compile all these records in a common database. Secondly, as this standard format offers new possibilities in terms of interaction with the data, we wanted to link this database to

an online portal. We estimated that the existing repositories were somehow limited in terms of interactivity, and that additionally to the new frame for datasets, it would be useful to provide new features for the sometimes fastidious online data browsing process, such as dynamic data browsing and visualization. This simultaneous effort to simplify the datasets format and improve data browsing makes the database and online portal highly interlinked. We estimated that this comment could be a good opportunity to clarify the original aim of this project in the article and therefore add some modifications to the abstract and introduction. We also agree that the original title of the article may have been somehow inappropriate and thus modified it as "Water and carbon stable isotope records from natural archives : a new database and interactive online platform for data browsing, visualizing and downloading".

Generalizability

We are favorable to the idea of sharing code and published data from our project. As the construction of the online portal is still ongoing, we cannot yet contribute to code repository, but will include it as soon as the version described in the article is fully functional and open to the community. As mentioned to Reviewer #1, we are also highly open to suggestions and collaboration with other institutions to upgrade our database and online portal.

Data standardization.

We think that LiPD data format might be a helpful improvement for data storage. However, we also think that a standard should be accepted by the whole community, and that all the data repositories should adopt this standard or a similar one. As we explain in a new paragraph of the manuscript, this format is a great opportunity to store and organize data from future publications. We however think that the fastest and less fastidious way to compile hundreds of previously published heterogeneous datasets is a single spreadsheet. This work was done in the frame of a post-doctoral fellowship so time and manpower were very limited to compile data and by the time of this data

compilation (2013 to late 2014), we were not aware of the existence of this LiPD format. Nevertheless, we encourage the community to adopt the LiPD as a new standard for metadata disposition, and we think that, if this format if finally accepted, our database could be relatively easily converted into such a structured standard. In the revised manuscript, we make explicit reference to this issue.

Data synthesis :

The LiPD standard and LinkedEarth project are indeed particularly promising. At the beginning of our work (2013), we were not aware of the existence of these projects and we adopted our own structure following what had been performed earlier for the MARGO project. The LiPD standard will highly facilitate future data storage with a comprehensive structure. We think that the original idea between the two concepts (metadata storage in a single spreadsheet versus individual files or tabs) may be different. LiPD is a great standard for future publications, as each author will be able to directly associate data and metadata in a hierarchical structure. However, considering the limited time and manpower, and as we had to compile and harmonize the existing datasets published along the last forty years, the single spreadsheet remained the fastest and most convenient solution for compiling metadata from these hundreds of datasets.

We also think that it would be feasible, with the participation of the community, to extract the information contained in our metadata spreadsheet and convert it to a structured format, and the LiPD seems particularly appropriate for that. Note that the time required to fill in the LiPD is approximately 10 minute per entry, and therefore the conversion is a workload way beyond our internal capabilities, and will require a community effort, as currently ongoing with the PAGES2k project. As we have also exchanged metadata information with the ISO2k project, we expect that the most recent records from our data base will be soon also available in the LiPD format. We therefore added a paragraph in the manuscript to highlight the need for the community to validate a definitive and interoperable format for metadata before starting to convert all the information from the

compiled datasets.

**Age modelling**

The original aim of this work performed in the frame of a post-doctoral position was the conversion of hundreds of heterogeneously formatted (age format, file disposition,…) datasets into homogenous records, so we had a very limited time and manpower to additionally gather and compile age-model information, as these data are particularly fragmented. We however agree that the next step in paleoclimate data formatting should be focused on age model information. Similarly to what we mentioned for the metadata, an agreement on the contents and format disposition has to be found within the paleoclimate community, before starting compiling data, otherwise numerous new "standards" types of containers will emerge and the problem of homogeneity will persist. Also, as mentioned by reviewer #2, age model precision and uncertainties became more and more crucial during last years, particularly for the study of fast and abrupt climatic events and transitions. Consequently, it is more and more important to gather all parameters used for the establishment of age models, as well as their associated uncertainties. Unfortunately, many of the records we collected were published more than twenty years ago, and the associated information concerning age model establishment is very limited and incomplete, notably concerning uncertainties.

**Chronology ratings**

Rating age models involves both qualitative and quantitative factors, and although we did our best to find a clear rating procedure, we thus agree that the expert judgment associated with qualitative aspects might inevitably be challenged. We believe that evaluating qualitative information on measurements such as the posterior distribution of ages is a constructive idea. We also think that asking the authors to provide this information with their future publications will contribute to enhance the evaluation of age models, but we also consider that calculating this distribution for the hundreds of published datasets might necessitate considerable time and manpower. We would be

favorable to include this information in our database and evaluation of the age models, but we would definitively need a collective effort to perform this task.

Statistical analysis

The significance of the difference between two different PMIP time slices was assessed by simply comparing the offset between the average isotopic value of these two periods, to the average value of the standard deviations of the isotopic record for each of the two periods.

We consider that the isotopic offset is (not) significant if the absolute value of the offset is greater (smaller) than the average standard deviation along the two periods. This information is now provided in the appendix of the revised manuscript.

Editorial Comments

We modified the manuscript and figures according to the constructive comments of reviewer #2. We however tried to modify the figures in A1 to A4 by making classes of number of datapoints but we estimate that this treatment leads to a loss of information and thus decided to keep the original figures as they were submitted although we agree that some of them might look a bit spiky.